# Self-supervised Meta-Prompt Learning with Meta-Gradient Regularization for Few-shot Generalization

**Kaihang Pan**[1,2*]**, Juncheng Li**[1†]**, Hongye Song**[2]**, Jun Lin**[2]**, Xiaozhong Liu**[3]**, Siliang Tang**[1]
[1] Zhejiang University, [2] DAMO Academy, Alibaba Group,
[3] Worcester Polytechnic Institute
{kaihangpan, junchengli, siliang}@zju.edu.cn
{hongye.shy, linjun.lj}@alibaba-inc.com, xliu14@wpi.edu

## Abstract

Prompt tuning is a parameter-efficient method, which learns soft prompts and conditions frozen language models to perform specific downstream tasks. Though effective, prompt tuning under few-shot settings on the one hand heavily relies on a good initialization of soft prompts. On the other hand, it can easily overfit to few-shot training samples, thereby undermining generalizability. Existing works leverage pre-training or supervised meta-learning to initialize soft prompts but they fail to data-efficiently generalize to unseen downstream tasks. To address the above problems, this paper proposes a novel **S**elf-s**U**pervised meta-**P**rompt learning framework with **ME**ta-gradient **R**egularization for few-shot generalization (**SUPMER**). SUPMER leverages self-supervised meta-learning with a diverse set of well-designed meta-training tasks to learn a universal prompt initialization for efficient adaptation using only unlabeled data. Additionally, it jointly meta-learns a gradient regularization function to transform raw gradients into a domain-generalizable direction, thus alleviating the problem of overfitting. Extensive experiments show that SUPMER achieves better performance for different few-shot downstream tasks, and also exhibits a stronger domain generalization ability. The code for SUPMER will be available at https://github.com/beepkh/SUPMER.

## 1 Introduction

Recent NLP accomplishments witnessed the rapid development of pre-trained language models (PLMs) (*e.g.*, BERT Devlin et al., 2019; T5 Raffel et al., 2020; GPT3 Brown et al., 2020). Fine-tuning, which tunes the entire PLM parameters, has achieved outstanding performances in various NLP tasks. However, as the pre-trained model scale increases, tuning the entire set of parameters

* Work done when interning at Alibaba DAMO Academy.
† Corresponding Author.

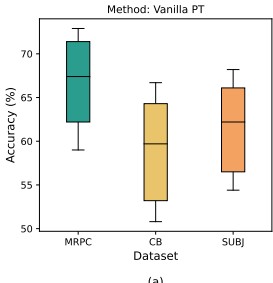
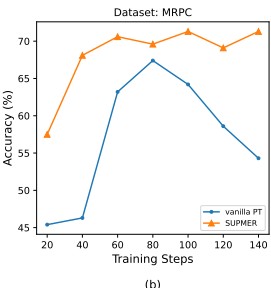

Figure 1: (a) Performance of PT with different prompt initialization. (b) Performance after different training steps for vanilla PT and SUPMER.

would be sometimes unaffordable. More recently, prompt-based methods, which simply insert a piece of carefully designed text to the input (*e.g.*, "*It was* ⟨*X*⟩.") and predict target words (*e.g.*, "great" or "terrible") at the mask position with frozen PLMs, have demonstrated remarkable effectiveness. But it has been observed that the performance of prompt-based methods is greatly affected by the design of prompts. In light of this, prompt tuning (PT Lester et al., 2021), as a parameter-efficient tuning method, is proposed to only prepend some additional learnable tokens called soft prompts to the input text, with all PLM parameters freezing.

Though prompt tuning is an efficient and effective paradigm, Gu et al. (2022) shows it performs much worse than fine-tuning under few-shot settings. We argue that the performance is not satisfactory mainly due to two limitations: 1) The performance of PT is highly **sensitive to the soft prompt initialization**, especially for few-shot tasks. As shown in Figure 1 (a), different soft prompt initialization leads to significant performance variations. 2) Few-shot PT risks **overfitting to some spurious correlations** as soft prompts are tuned on limited training samples, thus undermining the generalizability of PLMs. As shown in Figure 1 (b), the performance of few-shot vanilla PT degrades significantly in the final training steps.

Recent research mainly focused on the first limi-

tation, leveraging pre-training or supervised meta-learning for soft prompt initialization. A pre-trained prompt tuning method (PPT) (Gu et al., 2022) is proposed from the beginning, which utilizes self-supervised tasks to pre-train soft prompts and then applies them in the few-shot scenario. However, without explicitly optimizing the fast adaptation ability of the model, PPT suffers from a train-test mismatch between the pre-training data and the downstream data. So it limits generalization to unseen few-shot tasks, especially when there is a significant disparity in task domains or formats. MetaPrompting (Hou et al., 2022), as another effort, seeks assistance from model-agnostic meta-learning (MAML Finn et al., 2017) for fast adaptation in few-shot settings. However, in each task, MetaPrompting requires plenty of labeled data within certain classes to perform supervised meta-learning for prompt initialization, which is often inaccessible in practical few-shot scenarios. And the learned initialization can only generalize to the remaining classes of the same task in a few-shot manner, exhibiting weak task transferability. Furthermore, all these existing works ignore the second limitation, *i.e.*, the propensity for few-shot prompt tuning to lead to overfitting.

To address the shortcomings of existing works, we propose **SUPMER**, a **S**elf-s**U**pervised meta-**P**rompt learning framework with **ME**ta-gradient **R**egularization. It leverages self-supervised meta-learning to universally learn an efficient soft prompt initialization, also with a meta-gradient regularization function to mitigate overfitting. This comprehensive process only ***requires a one-time execution*** and enables seamless adaptation to different downstream few-shot tasks, while also facilitating faster convergence for downstream prompt tuning.

Specifically, to address the first limitation, we design a novel self-supervised meta-learning method for prompt initialization, which automatically generates a diverse set of meta-training tasks from large-scale unlabeled corpora and explicitly learns to fast adapt across these tasks. To ensure task diversity, we initially design a collection of anchor self-supervised meta-training tasks with different formats. And then a curriculum-based task augmentation method is further proposed to enrich the task distribution dynamically in terms of the current model capability.

For the second issue, we integrate a meta-gradient regularization function into meta-prompt

learning. As we simulate distribution shift through task augmentation, the meta-gradient regularization parameters are jointly optimized to align gradient directions across different distributions within our proposed meta-prompt learning paradigm. Consequently, in downstream tasks, these optimized parameters can be directly utilized to transform raw gradients over few-shot samples into a domain-generalizable direction, preventing prompt tuning overfitting to some domain-specific correlations.

Overall, our contributions are mainly three-fold:

(1) We propose a novel self-supervised meta-prompt learning framework to better initialize soft prompts, where only unlabeled pre-training data are used to construct different meta-training tasks with curriculum-based task augmentation for further task enrichment.

(2) We incorporate a novel meta-gradient regularization function into our meta-prompt learning framework, which meta-learns to transform the raw gradient during few-shot learning into a domain-generalizable direction, thus preventing prompt tuning overfitting to domain-specific correlations.

(3) Comprehensive experiments on few-shot learning and domain generalization validate the superiority of our method, which even outperforms full-model tuning in few-shot learning. It also exhibits a stronger domain generalization ability.

## 2 Related Work

**Soft Prompt Tuning.** Soft prompt tuning is one of the most parameter-efficient tuning methods widely used in NLP (Liu et al., 2023) and vision-language tasks (Zhou et al., 2022; Li et al., 2023a), which only tunes a small number of (extra) parameters to attain strong performance. Specifically, it freezes the PLM parameters and prepends some trainable continuous embeddings (*i.e.*, soft prompts) to the input sequence (Lester et al., 2021) or every layer of the pre-trained model (Li and Liang, 2021; Liu et al., 2022).

To efficiently train task-adaptive soft prompts in few-shot scenarios, some studies (Vu et al., 2022; Asai et al., 2022; Sun et al., 2022) employ task adaptation techniques, obtaining source prompts from source tasks in a supervised way and interpolating them into the target prompts. Other works focus on training improved prompt initializations. PPT (Gu et al., 2022) pre-trains the soft prompts with some self-supervised tasks on unlabeled corpora, but it doesn't explicitly optimize the fast adap-

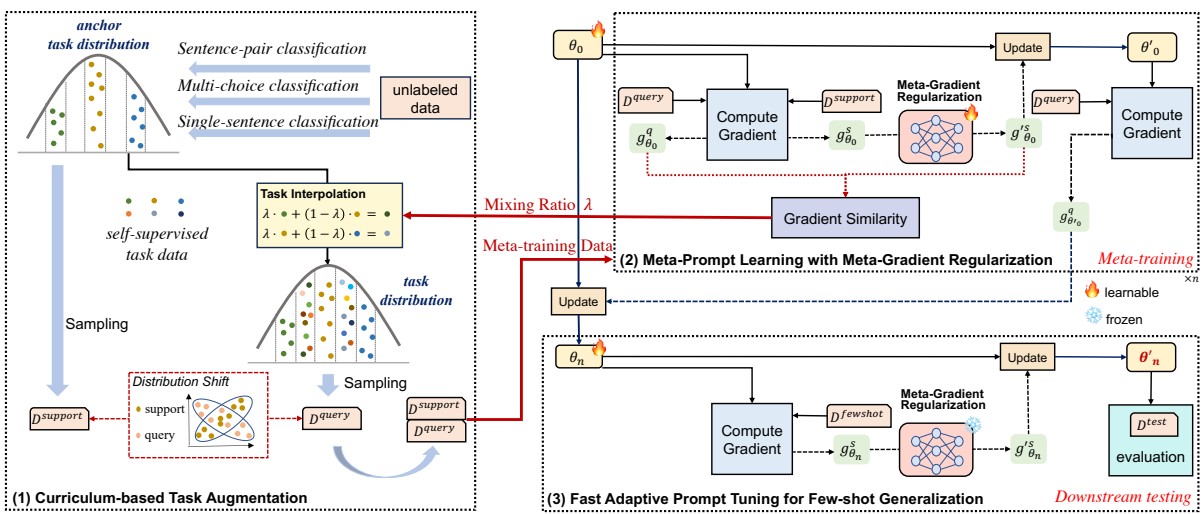

Figure 2: The framework of SUPMER. We employ task interpolation to enrich the distribution of self-supervised meta-training tasks. Concurrently, we integrate a meta-gradient regularization function into meta-prompt learning. Furthermore, during meta-prompt learning we also dynamically adapt the mixing ratio of task interpolation, upgrading the vanilla task augmentation into a curriculum-based one.

tation ability of the model. MetaPrompting(Hou et al., 2022) utilizes supervised meta-learning for soft prompt initialization, splitting each dataset into two sets with disjoint data classes. One split is used to initialize soft prompts while the other serves as the downstream task. In comparison, SUPMER differs from MetaPrompting in the following ways: 1) for each downstream task MetaPrompting focuses on a fixed supervised dataset to reinitialize soft prompts, whereas SUPMER can universally generalize to different unseen tasks with large-scale unlabeled corpora for initialization; 2) MetaPrompting doesn't freeze PLM parameters, while SUPMER only tunes the soft prompts as the general soft prompt tuning methods do.

**Meta-Learning.** Meta-learning, also known as learning to learn, optimizes the ability to learn new tasks quickly and efficiently, utilizing experience from previously seen tasks. It can be classified into three types: metric-based methods (Koch et al., 2015; Vinyals et al., 2016; Snell et al., 2017), model-based methods (Graves et al., 2014; Mishra et al., 2018; Qiao et al., 2018), and gradient-based methods (Hochreiter et al., 2001; Ravi and Larochelle, 2017; Nichol et al., 2018; Li et al., 2020). In this work, we focus on a gradient-based meta-learning algorithm (*i.e.*, MAML Finn et al., 2017). Compared to typical meta-learning methods that rely on human-annotated meta-training tasks, we automatically generate abundant tasks in a self-supervised way, also integrating a meta-gradient regularization function into MAML to steer gradi-

ents towards a domain-generalizable direction.

## 3 Method

In this section, we describe the whole framework of SUPMER (shown in Figure 2). With pre-defined preliminaries, we first introduce the way to construct anchor self-supervised meta tasks and the foundation of task augmentation to densify task distributions. Then we elaborate on the SUPMER model, including the meta-gradient regularization function. Finally, we upgrade the original task augmentation method into a curriculum-based one. Besides, we formalize all tasks in a text-to-text format following the T5 fashion (Raffel et al., 2020).

### 3.1 Preliminaries

**Prompt Tuning.** In prompt tuning (Lester et al., 2021), given a training sample $(x_i, y_i)$ from task $\mathcal{D}_\tau$, we apply a prompt template $P$ converting $x_i$ into a new sequence $P(x_i)$ and then concatenate a set of soft prompts $\theta$ to the beginning of $P(x_i)$. And verbalizer $\mathcal{V}$ plays a role in mapping $y_i$ to some corresponding label tokens $\mathcal{V}(y_i)$ in the vocabulary of PLMs. So the objective of prompt tuning can be formulated as follows:

$$\arg\min_\theta \mathcal{L}_{\mathcal{D}_\tau}(\theta)$$
$$= \arg\max_\theta \sum_{(x_i,y_i)\in D_\tau} \log p\big(\langle X\rangle = \mathcal{V}(y_i)|[\theta; P(x_i)]; \theta\big)$$
$$(1)$$

where $\theta$ denotes the soft prompt embedding (the only tunable parameters in prompt tuning). $\langle X\rangle$ let PLMs predict target tokens at the masked positions and $[\cdot; \cdot]$ is the concatenation operation.

**Model-Agnostic Meta-Learning.** Assuming access to a task distribution $p(\mathcal{T})$, the goal of meta-learning is to utilize tasks $\tau_i \sim p(\mathcal{T})$, referred to as meta-training tasks or meta tasks, to train a learning procedure that generalizes to unseen tasks from the distribution. Model-Agnostic Meta-Learning (MAML) (Finn et al., 2017) is a gradient-based bi-level optimization meta-learning method, which consists of an inner loop task-specific learning and outer loop fast adaptation across tasks.

Specifically, a task $\tau$ is composed of the support set $\mathcal{D}_\tau^s$ and the query set $\mathcal{D}_\tau^q$. In the inner loop of MAML, a model learns to adapt to a new task $\tau_i$ using its support set in the following way:

$$\theta_i' = \theta - \alpha_1 \nabla_\theta \mathcal{L}_{\mathcal{D}_{\tau_i}^s}(\theta) \tag{2}$$

where $\alpha_1$ is the inner loop learning rate and $\theta$ is the model's parameters. And the optimized parameters $\theta_i'$ is then evaluated on the query set of task $\tau_i$ with the loss function $\mathcal{L}_{\mathcal{D}_{\tau_i}^q}$. In the outer loop, this loss across meta-training tasks is treated as the final training loss to update $\theta$:

$$\theta \leftarrow \theta - \beta_1 \nabla_\theta \sum_{\tau_i \sim p(\mathcal{T})} \mathcal{L}_{\mathcal{D}_{\tau_i}^q}(\theta_i') \tag{3}$$

where $\beta_1$ is the outer loop learning rate.

### 3.2 Constructing Anchor Meta Tasks

Supervised datasets with a large amount of labeled data are often unavailable in many NLP tasks. While unlabeled data is more easily accessible and generally covers broader semantic concepts. So we utilize the unlabeled data from a large corpus to create anchor self-supervised meta-training tasks.

The unlabeled data are first grouped into different clusters. We utilize PLMs to derive semantically meaningful embeddings for sentences in the corpus, and then apply unsupervised K-means to cluster these unlabeled sentences. Based on the results of K-means, we design three different formats of self-supervised meta-training tasks: sentence-pair classification, multi-choice classification, and single-sentence classification.

Specifically, **sentence-pair classification** involves predicting whether two sentences are adjacent in the same document or from the same cluster after K-means clustering. **Multi-choice classification** identifies the correct sentence among several candidates, which is either adjacent to a query sentence or from its same cluster. And **Single-sentence classification** aims to associate each sentence with its correct cluster label, as determined

by K-means. On this basis, for each task format, we distribute meta-training data into different tasks to construct anchor meta-training tasks with well-balanced task distributions. We group samples with similar embeddings into the same task based on the results of K-means. And we give a more detailed description of anchor meta-training task construction in Appendix A.2.

### 3.3 Vanilla Task Augmentation

With a set of anchor meta-training tasks, in this section we first introduce the vanilla task augmentation to densify the task distribution. Extending the idea of mixup (Zhang et al., 2018), we augment the task set through task interpolation, which linearly combines features and corresponding labels of samples from the query set in different tasks. In §3.5 we further upgrade the vanilla task augmentation method into a curriculum-based one, which dynamically controls the task interpolation in terms of the current model capability.

Specifically, for a task composed of the support set and the query set, we denote the hidden representations of the query set samples in task $\tau_k$ as $\boldsymbol{H}^q$. Given an anchor task $\tau_i$, first we randomly select another task $\tau_j$. While retaining the support set of $\tau_i$, we reconstruct its query set by interpolating on the hidden representations $(\boldsymbol{H}_i^q, \boldsymbol{H}_j^q)$ and corresponding labels $(\boldsymbol{Y}_i^q, \boldsymbol{Y}_j^q)$ from the query sets in $\tau_i$ and $\tau_j$, which can be accomplished using mixup:

$$\tilde{\boldsymbol{H}}_i^q = (1-\lambda)\boldsymbol{H}_i^q + \lambda\boldsymbol{H}_j^q, \ \tilde{\boldsymbol{Y}}_i^q = (1-\lambda)\boldsymbol{Y}_i^q + \lambda\boldsymbol{Y}_j^q \tag{4}$$

where the mixing ratio $\lambda \in [0, 1]$ is drawn from a Beta distribution $Beta(\alpha, \alpha)$, and $\alpha$ is a hyper-parameter. The process of task augmentation not only enriches the task distribution, but also simulates the distribution shift between the support set and the query set within one task, as we only leverage interpolation between the query sets of different anchor meta-training tasks. And in §3.4 we will show the effect of this distribution deviation.

### 3.4 Meta-Prompt Learning with Meta-Gradient Regularization

In this section we introduce the algorithm of our meta-prompt learning framework, which is a bi-level meta-learning paradigm learning a task-universal soft prompt initialization $\theta$ for efficient adaptation. And it jointly meta-learns a meta-gradient regularization function $\psi_\phi$ that transforms raw gradients into a domain-generalizable direction to prevent prompt tuning from overfitting.

Specifically, considering that the inner loop update of MAML (*i.e.*, Eq. (2)) over limited samples might overfit to some domain-specific correlations, we propose to learn a gradient regularization function $\psi_\phi(\cdot)$, making a direct transformation to the raw gradients obtained from the support set $\mathcal{D}^s_{\tau_i}$. The function first performs affine transformation $h(\cdot)$ (*e.g.*, rotation) to modulate the raw gradients $g$, and then an update gate vector $z$ is employed to combine $g$ and $h(g)$ into the final gradients:

$$\psi_\phi(g) = z \cdot h(g) + (1 - z) \cdot g \qquad (5)$$

Obviously, the value of $z$ can be used to control how much the transformed gradients $h(g)$ contribute to the output of $\psi_\phi(g)$. We hope to determine this weight based on the input samples themselves, setting $z = \sigma(W\boldsymbol{H} + b)$, where $\boldsymbol{H}$ is the hidden representations of input samples. Formally, now we transform Eq. (2) into:

$$\theta'_i = \theta - \alpha_1 \psi_\phi(\nabla_\theta \mathcal{L}_{\mathcal{D}^s_{\tau_i}}(\theta)) \qquad (6)$$

After adapting the soft prompt embeddings to the support set $\mathcal{D}^s_{\tau_i}$, in the outer loop we optimize the prompt initialization $\theta$ based on these adapted embeddings $\theta'$ via Eq. (3). Besides, meta-gradient regularization parameters $\phi$ are also optimized using the same loss to learn a better gradient transformation, with $\beta_2$ as the learning rate:

$$\phi \leftarrow \phi - \beta_2 \nabla_\phi \sum_{\tau_i \sim p(\mathcal{T})} \mathcal{L}_{\mathcal{D}^q_{\tau_i}}(\theta'_i) \qquad (7)$$

Overall, the total meta-prompt learning obejective can be formulated as follows:

$$\arg\min_{\theta,\phi} \sum_{\tau_i \sim p(\mathcal{T})} \mathcal{L}_{\mathcal{D}^q_{\tau_i}} \big( \theta - \alpha_1 \psi_\phi(\nabla_\theta \mathcal{L}_{\mathcal{D}^s_{\tau_i}}(\theta)) \big) \quad (8)$$

**Downstream Prompt Tuning.** The above meta-prompt learning framework only requires a one-time execution. *The optimized prompt initialization $\theta^*$ and meta-gradient regularization parameters $\phi^*$ are then universal for different downstream tasks.* During downstream prompt tuning, we fix $\phi^*$ and further adapt $\theta^*$ to testing tasks as Eq. (6).

**Analysis of SUPMER.** Here we give some analysis of how SUPMER could enhance generalizability, with more complete proof in Appendix A.1. Given that $x = \theta - \alpha_1 \psi_\phi(\nabla_\theta \mathcal{L}_{\mathcal{D}^s}(\theta))$ and $x_0 = \theta$, focusing on a single meta-training task, we can apply a first-order Taylor expansion around the point

$x_0$ to reformulate Eq. (8) as:

$$\because \mathcal{L}_{\mathcal{D}^q}(x) = \mathcal{L}_{\mathcal{D}^q}(x_0) + \mathcal{L}'_{\mathcal{D}^q}(x_0)(x - x_0)$$
$$\therefore \arg\min_{\theta,\phi} \mathcal{L}_{\mathcal{D}^q} \big( \theta - \alpha_1 \psi_\phi(\nabla_\theta \mathcal{L}_{\mathcal{D}^s}(\theta)) \big) \qquad (9)$$
$$= \arg\min_{\theta,\phi} \mathcal{L}_{\mathcal{D}^q}(\theta) - \alpha_1 \nabla_\theta \mathcal{L}_{\mathcal{D}^q}(\theta) \cdot \psi_\phi\big(\nabla_\theta \mathcal{L}_{\mathcal{D}^s}(\theta)\big)$$

Based on the aforementioned discussion, we can reach the following conclusions: (1) The update of $\theta$ minimizes the expected loss on the query set. (2) The optimization of both $\theta$ and $\phi$ maximizes the inner product between the regulated gradients from the support set and the gradients from the query set. The inner product of two vectors is larger if they are in a similar direction. Recalling that we simulate the distribution shift between the support set and the query set, the optimization of $\theta$ and $\phi$ tries to align the gradient directions across different distributions. To improve the alignment between the domain-specific gradients, the gradient regularization parameters $\phi$ are optimized to retain some domain-invariant information of meta-training data and then can be utilized to regulate raw gradients obtained from few-shot samples into a domain-generalizable direction in downstream prompt tuning, thus avoiding overfitting to some spurious correlations.

### 3.5 Curriculum-based Task Augmentation

In §3.4 we show that SUPMER can help align the optimization direction across two distributions with deviation, which is simulated by performing task augmentation exclusively on the support sets. From Eq. (4) it is evident that the mixing ratio $\lambda$ of mixup controls the extent of the distribution deviation, with a larger $\lambda$ resulting in a more noticeable deviation. However, in the previously discussed method, $\lambda$ is sampled from a fixed Beta distribution. In this section, we propose a more flexible sampling approach, which upgrades the original task augmentation method into a curriculum-based one, gradually increasing the task difficulty and achieving a more reasonable distribution shift.

The curriculum-based task augmentation dynamically adjusts the parameters of the Beta distribution, from which we sample the mixing ratio $\lambda$. Specifically, a batch of meta tasks is sampled in each training epoch. For each task, we can obtain gradients on the support set $g^s_i$ and gradients on the query set $g^q_i$, along with their cosine similarity. We leverage the average cosine similarity $s_{k-1}$ of all tasks in a batch during the last epoch to derive the

| Methods | SST-2 | SST-5 | MR | CR | SUBJ | TREC | CB | RTE | QNLI | WiC | MRPC | QQP | AVG |
|---|---|---|---|---|---|---|---|---|---|---|---|---|---|
| **Model: T5-base (220M)** | | | | | | | | | | | | | |
| Prefix Tuning | $78.2_{1.7}$ | $38.0_{2.5}$ | $74.0_{3.2}$ | $84.5_{3.3}$ | $66.2_{3.9}$ | $70.1_{2.9}$ | $70.4_{2.6}$ | $54.4_{2.0}$ | $55.7_{4.3}$ | $53.4_{1.3}$ | $68.1_{1.2}$ | $64.9_{1.3}$ | $64.8_{2.5}$ |
| P-Tuning-v2 | $83.1_{0.8}$ | $41.7_{1.5}$ | $82.3_{0.8}$ | $88.7_{0.5}$ | $74.8_{2.4}$ | $78.1_{2.1}$ | $67.5_{5.4}$ | $53.9_{1.3}$ | $61.4_{1.6}$ | $54.5_{2.2}$ | $70.7_{1.1}$ | $68.0_{1.1}$ | $68.7_{1.7}$ |
| FT | $83.6_{1.7}$ | $41.2_{2.6}$ | $81.7_{0.9}$ | $88.3_{0.9}$ | $80.0_{1.4}$ | $79.8_{1.7}$ | $71.9_{1.5}$ | $56.9_{2.1}$ | $\mathbf{62.3_{0.6}}$ | $54.6_{1.6}$ | $70.2_{0.7}$ | $69.5_{1.0}$ | $70.0_{1.4}$ |
| PT | $71.9_{3.4}$ | $37.3_{2.8}$ | $73.2_{4.5}$ | $84.4_{3.5}$ | $61.5_{5.3}$ | $65.3_{4.2}$ | $58.9_{6.1}$ | $53.2_{2.8}$ | $55.2_{4.8}$ | $53.1_{2.4}$ | $66.6_{5.3}$ | $63.0_{2.6}$ | $62.0_{4.0}$ |
| PPT | $81.2_{2.0}$ | $40.2_{5.4}$ | $81.2_{0.7}$ | $83.6_{7.3}$ | $66.8_{3.7}$ | $73.4_{2.4}$ | $60.7_{7.7}$ | $55.4_{1.2}$ | $60.4_{3.9}$ | $53.6_{1.3}$ | $68.0_{0.8}$ | $63.1_{0.7}$ | $65.6_{3.1}$ |
| Unified-PPT | $76.8_{7.7}$ | $44.7_{1.7}$ | $79.0_{3.3}$ | $87.7_{0.6}$ | $64.2_{5.8}$ | $68.4_{2.9}$ | $65.4_{2.4}$ | $54.4_{1.1}$ | $56.7_{2.4}$ | $54.5_{1.6}$ | $67.8_{1.1}$ | $67.6_{2.9}$ | $65.5_{2.8}$ |
| MetaPT | $85.7_{1.3}$ | $45.3_{1.1}$ | $82.5_{4.5}$ | $88.5_{0.3}$ | $73.2_{3.7}$ | $78.7_{2.2}$ | $65.4_{2.4}$ | $56.1_{1.7}$ | $58.2_{3.1}$ | $54.1_{1.3}$ | $69.6_{0.6}$ | $68.8_{0.9}$ | $68.8_{1.9}$ |
| **SUPMER** | $\mathbf{87.3_{0.5}}$ | $\mathbf{46.7_{0.6}}$ | $\mathbf{84.0_{0.6}}$ | $\mathbf{89.3_{0.3}}$ | $79.6_{2.2}$ | $\mathbf{80.2_{0.9}}$ | $\mathbf{72.4_{1.4}}$ | $\mathbf{57.3_{1.0}}$ | $61.7_{1.0}$ | $\mathbf{54.8_{1.2}}$ | $\mathbf{71.3_{0.5}}$ | $\mathbf{70.5_{1.0}}$ | $\mathbf{71.3_{0.9}}$ |
| **Model: Flan-T5-XL (3B)** | | | | | | | | | | | | | |
| Zero-shot Inference | 89.1 | 52.3 | 83.3 | 80.6 | 57.4 | 87.2 | 76.8 | 75.8 | 85.0 | 50.5 | 77.2 | 77.5 | 74.4 |
| Few-shot Inference | 93.2 | 53.3 | 88.5 | 87.8 | 58.6 | 91.6 | 83.9 | 79.1 | 86.9 | 64.3 | **79.9** | 81.0 | 79.0 |
| Prefix Tuning | $88.6_{2.1}$ | $45.4_{2.7}$ | $88.7_{2.2}$ | $89.1_{1.6}$ | $84.6_{3.9}$ | $90.6_{2.4}$ | $65.4_{6.4}$ | $73.4_{2.6}$ | $72.3_{4.1}$ | $55.5_{1.8}$ | $73.9_{2.6}$ | $75.0_{2.3}$ | $75.2_{2.9}$ |
| P-Tuning-v2 | $91.2_{1.0}$ | $53.5_{1.1}$ | $90.0_{0.8}$ | $89.8_{1.1}$ | $87.7_{3.2}$ | $92.3_{0.9}$ | $69.1_{2.9}$ | $62.3_{4.6}$ | $76.8_{3.6}$ | $63.1_{2.0}$ | $77.2_{1.7}$ | $77.3_{1.6}$ | $77.5_{2.0}$ |
| FT | $92.9_{1.4}$ | $53.6_{1.9}$ | $89.6_{1.3}$ | $\mathbf{91.3_{0.5}}$ | $88.7_{0.9}$ | $\mathbf{93.6_{2.0}}$ | $77.8_{1.6}$ | $76.0_{1.7}$ | $84.4_{1.5}$ | $63.0_{1.8}$ | $77.7_{1.3}$ | $82.9_{2.9}$ | $81.0_{1.6}$ |
| PT | $88.2_{9.2}$ | $45.7_{3.4}$ | $85.6_{5.6}$ | $88.6_{5.1}$ | $81.6_{5.1}$ | $85.1_{5.6}$ | $65.4_{7.2}$ | $61.6_{4.0}$ | $69.3_{5.9}$ | $54.8_{1.1}$ | $71.1_{2.7}$ | $70.2_{8.6}$ | $72.3_{5.3}$ |
| PPT | $91.0_{2.6}$ | $49.1_{2.3}$ | $88.8_{1.3}$ | $88.0_{3.0}$ | $87.6_{1.9}$ | $90.8_{1.7}$ | $67.1_{4.8}$ | $67.1_{2.2}$ | $81.5_{1.1}$ | $56.1_{1.3}$ | $75.8_{2.2}$ | $71.6_{3.1}$ | $76.2_{2.3}$ |
| Unified-PPT | $89.9_{8.3}$ | $47.4_{2.8}$ | $89.7_{0.9}$ | $89.0_{1.5}$ | $86.4_{4.3}$ | $88.4_{3.2}$ | $69.3_{3.5}$ | $63.2_{1.1}$ | $74.9_{3.2}$ | $59.9_{1.7}$ | $74.0_{2.4}$ | $76.3_{2.0}$ | $75.7_{2.9}$ |
| MetaPT | $93.7_{0.8}$ | $52.5_{1.3}$ | $90.7_{0.7}$ | $89.7_{0.6}$ | $88.1_{2.4}$ | $91.8_{1.8}$ | $72.1_{2.2}$ | $74.1_{1.3}$ | $77.5_{2.5}$ | $58.5_{1.6}$ | $76.6_{1.8}$ | $81.4_{2.4}$ | $78.9_{1.6}$ |
| **SUPMER** | $\mathbf{95.5_{0.4}}$ | $\mathbf{55.3_{0.7}}$ | $\mathbf{91.4_{0.5}}$ | $90.7_{0.7}$ | $\mathbf{90.3_{0.8}}$ | $93.0_{1.5}$ | $\mathbf{87.6_{1.5}}$ | $\mathbf{81.4_{1.0}}$ | $\mathbf{88.3_{0.6}}$ | $\mathbf{65.0_{1.7}}$ | $78.1_{0.8}$ | $\mathbf{85.1_{0.4}}$ | $\mathbf{83.5_{0.9}}$ |

Table 1: Results of few-shot learning. For each dataset we report the average accuracy and standard deviation over five random seeds (zero-shot & few-shot inference produce nearly consistent results each time as they do not require parameter tuning). **Bold fonts** indicate the best results. We can see SUPMER achieves better performance.

mixing ratio $\lambda_k$ for the current epoch $k$:

$$\lambda_k = Beta(\alpha, b_k\alpha)$$

$$b_k = \frac{m^{\frac{1+s_{k-1}}{2}} - 1}{m - 1},$$ (10)

$$\text{where } s_{k-1} = \frac{1}{|\mathcal{B}|} \cdot \sum_{i=1}^{|\mathcal{B}|} \frac{g_i^s \cdot g_i^q}{\|g_i^s\| \cdot \|g_i^q\|}$$

where $m$ is the curve parameter. In this way, when our model is not capable of aligning the optimization directions across different distributions at the beginning, a smaller $\lambda$ is preferable to create a smaller distribution deviation. Then $\lambda$ tends to gradually increase as the model's capability improves, resulting in a larger distribution deviation and a corresponding increase in task difficulty.

We present the pseudo-codes of SUPMER in Appendix A.4.

## 4 Experiments

### 4.1 Experimental Setup

We evaluate our approach in two problem settings: 1) Few-shot learning with different NLP downstream tasks; 2) domain generalization.

**Few-shot Learning.** We consider 6 downstream tasks with 12 datasets: 1) the sentiment analysis datasets SST-2, SST-5 (Socher et al., 2013), MR (Pang and Lee, 2005) and CR (Hu and Liu, 2004); 2) the subjectivity classification dataset SUBJ (Pang and Lee, 2004); 3) the question classification dataset TREC (Voorhees and Tice, 2000); 4) the natural language inference datasets CB (De Marneffe et al., 2019) and RTE (Wang et al., 2019); 5) the question answering dataset QNLI (Rajpurkar et al., 2016); 6) the word sense disambiguation dataset WiC (Pilehvar and Camacho-Collados, 2019); 7) the paraphrase detection datasets MRPC (Dolan and Brockett, 2005) and QQP. Following Karimi Mahabadi et al. (2022), for each dataset we sample 16 instances per label from the original training set to form training and validation sets for few-shot learning.

**Domain Generalization.** Then we design a more challenging problem about zero-shot domain generalization in the sentiment analysis task. Our experiments include 6 domains across 3 datasets: 1) the Amazon review dataset (Blitzer et al., 2007) containing reviews about Books (B), DVDs (D), Electronics (E) and Kitchen appliances (K); 2) the airline review dataset (A) (Nguyen, 2015; Ziser and Reichart, 2018); 3) the restaurant (R) domain obtained from the Yelp dataset (Zhang et al., 2015).

We choose A as the source domain and the other five (B, D, E, K, R) constitute the target domains. On this basis, we sample 16 instances per label from the training set of the source domain to tune soft prompts. And then we directly use the soft prompts learned from the source domain to evaluate performance on the test set of each domain.

| | Model: T5-base (220M) | | | | | | |
|---|---|---|---|---|---|---|---|
| **Method** | **Source** | **Target** | | | | | **AVG** |
| | **A** | **B** | **D** | **E** | **K** | **R** | |
| Prefix Tuning | $78.1_{2.2}$ | $82.7_{1.0}$ | $81.5_{1.2}$ | $84.4_{0.6}$ | $82.7_{2.6}$ | $84.9_{2.2}$ | $82.4_{1.6}$ |
| P-Tuning-v2 | $84.0_{0.8}$ | $83.6_{0.9}$ | $82.4_{1.7}$ | $85.9_{0.9}$ | $84.2_{1.5}$ | $83.8_{2.7}$ | $84.0_{1.4}$ |
| FT | $84.4_{0.2}$ | $83.9_{0.6}$ | $81.0_{0.6}$ | $84.1_{0.6}$ | $85.0_{0.7}$ | $85.3_{0.6}$ | $84.0_{0.6}$ |
| PT | $79.8_{2.5}$ | $75.3_{3.2}$ | $76.0_{3.2}$ | $79.6_{2.0}$ | $79.8_{1.9}$ | $83.0_{1.8}$ | $78.9_{2.3}$ |
| PPT | $81.9_{1.4}$ | $77.9_{3.5}$ | $83.6_{1.3}$ | $88.4_{1.0}$ | $89.1_{1.6}$ | $87.8_{1.5}$ | $84.8_{1.7}$ |
| Unified-PPT | $80.9_{3.0}$ | $82.1_{2.0}$ | $76.8_{4.0}$ | $81.0_{4.5}$ | $82.2_{3.9}$ | $84.9_{3.3}$ | $81.3_{3.5}$ |
| MetaPT | $\mathbf{86.1_{0.3}}$ | $84.3_{0.8}$ | $84.2_{0.7}$ | $89.4_{0.9}$ | $90.3_{0.6}$ | - | $(86.9_{0.7})$ |
| **SUPMER** | $85.7_{0.5}$ | $\mathbf{85.3_{0.4}}$ | $\mathbf{85.1_{0.4}}$ | $\mathbf{90.3_{0.6}}$ | $\mathbf{91.1_{0.5}}$ | $\mathbf{90.4_{0.4}}$ | $\mathbf{88.0_{0.5}}$ |
| | Model: Flan-T5-XL (3B) | | | | | | |
| **Method** | **Source** | **Target** | | | | | **AVG** |
| | **A** | **B** | **D** | **E** | **K** | **R** | |
| zero-shot inference | 77.8 | 84.6 | 86.2 | 86.8 | 88.6 | 87.8 | 85.3 |
| few-shot inference | 82.5 | 90.3 | 89.7 | 92.3 | 92.2 | 89.2 | 89.4 |
| Prefix Tuning | $83.0_{1.1}$ | $85.3_{1.9}$ | $83.4_{1.1}$ | $87.3_{1.5}$ | $86.4_{2.4}$ | $89.8_{1.0}$ | $85.9_{1.5}$ |
| P-Tuning-v2 | $85.6_{0.4}$ | $86.7_{2.3}$ | $86.8_{1.8}$ | $88.6_{2.1}$ | $90.2_{2.3}$ | $88.9_{2.6}$ | $87.8_{1.9}$ |
| FT | $86.2_{0.7}$ | $88.8_{2.2}$ | $84.6_{1.3}$ | $87.2_{1.3}$ | $89.8_{1.1}$ | $90.7_{0.9}$ | $87.9_{1.3}$ |
| PT | $82.6_{1.2}$ | $79.0_{3.9}$ | $82.5_{1.7}$ | $84.2_{2.3}$ | $84.5_{2.5}$ | $84.8_{2.1}$ | $82.9_{2.3}$ |
| PPT | $85.2_{0.9}$ | $83.2_{3.3}$ | $89.7_{1.8}$ | $92.4_{1.7}$ | $93.6_{1.5}$ | $89.6_{1.1}$ | $89.0_{1.7}$ |
| Unified-PPT | $83.0_{0.8}$ | $82.5_{3.0}$ | $82.0_{4.8}$ | $86.0_{3.0}$ | $86.6_{2.2}$ | $85.6_{2.0}$ | $84.3_{2.6}$ |
| MetaPT | $\mathbf{87.1_{0.6}}$ | $87.4_{2.7}$ | $89.5_{1.7}$ | $93.8_{0.6}$ | $94.3_{1.1}$ | - | $(90.4_{1.3})$ |
| **SUPMER** | $87.0_{0.2}$ | $\mathbf{89.8_{1.4}}$ | $\mathbf{91.1_{1.2}}$ | $\mathbf{95.1_{0.8}}$ | $\mathbf{95.8_{0.9}}$ | $\mathbf{91.8_{0.8}}$ | $\mathbf{91.8_{0.9}}$ |

Table 2: Results of domain generalization. For MetaPT we calculate the average performance only across domain A, B, D, E, K (without R).

## 4.2 Experimental Details

**Baselines.** Our experiments are built on a smaller-scale model, **T5-base** (Raffel et al., 2020), and then on a larger-scale instruction-tuned model, **Flan-T5-XL** (Chung et al., 2022). For both two backbone models, we use the following baselines: (1) prompt tuning methods with the same number of tunable parameters as SUPMER: vanilla prompt tuning (**PT** Lester et al., 2021), **PPT** (Gu et al., 2022), **Unified-PPT** (Gu et al., 2022), and **MetaPT** (Huang et al., 2022). (2) methods with more tunable parameters: **Prefix-Tuning** (Li and Liang, 2021), **P-tuning-v2** (Liu et al., 2022), full-model tuning (**FT**). Furthermore, Given that FLAN-T5-XL was also designed with few-shot inference in mind, we additionally compare with two baseline methods on FLAN-T5-XL, *i.e.,* **zero-shot inference** and **few-shot inference**, which directly employ Flan-T5-XL for downstream evaluation. We list the details of baselines in Appendix B.

**Implementation Details.** We solve all downstream tasks in a text-to-text format and run each experiment with 5 different random seeds. For all prompt tuning methods, we follow Lester et al. (2021) to design soft prompts composed of 100 soft tokens, with tunable parameters far less than full-model tuning. For our SUPMER, following PPT (Gu et al., 2022) we sample 10GB data from OpenWebText (Gokaslan et al., 2019), a large-scale unlabeled corpus, to construct self-supervised meta-training tasks. The meta-training stage only requires a one-time execution. In downstream prompt-tuning, we freeze the meta-gradient reg-

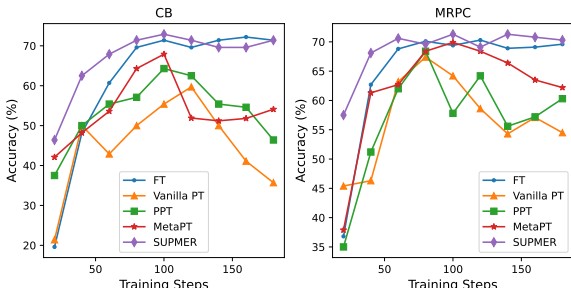

Figure 3: The performance after different training steps on CB and MRPC.

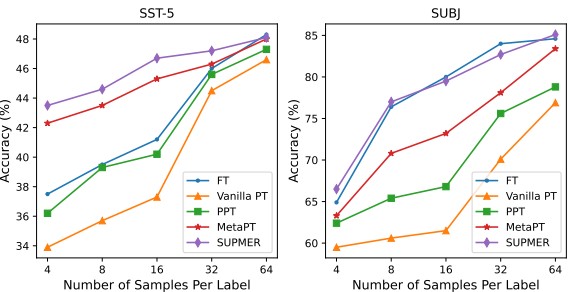

Figure 4: The performance on SST-5 and SUBJ when different numbers of training samples are available.

ularization parameters and the soft prompts are the only tunable parameters. We give more details of training hyper-parameters in Appendix C.

## 4.3 Main Result

Table 1 and Table 2 show the main results of few-shot learning and domain generalization. From the results, we have the following observations.

First, in few-shot learning, SUPMER achieves better performance than all baselines on 10 of 12 datasets, whether using T5-base or Flan-T5-XL as the backbone. And the average accuracy of SUPMER over all datasets reaches 71.3% on T5-base, significantly outperforming other baselines (*e.g.,* improving the performance by +1.3 points compared to FT). Notably, when utilizing the larger Flan-T5-XL as the backbone, SUPMER demonstrates even more substantial performance gains (*e.g.,* improving the average performance by +2.5 points compared to FT), which indicates that our approach unlocks greater capabilities for stronger models that have undergone instruction-tuning with a higher number of parameters.

Specifically, SUPMER consistently outperforms all other prompt tuning methods with the same number of tunable parameters across all datasets. This indicates that our method offers soft prompts with better few-shot generalization ability. And it is noteworthy to highlight that SUPMER utilizes ex-

| Methods | SST-2 | SST-5 | MR | CR | SUBJ | TREC | CB | RTE | QNLI | WiC | MRPC | QQP | **AVG** |
|---|---|---|---|---|---|---|---|---|---|---|---|---|---|
| 1 SUPMER (only labeled) | **87.5** | **47.0** | 83.8 | **89.9** | 75.4 | 79.6 | 67.9 | 56.6 | 59.0 | 54.6 | 69.2 | 69.5 | 70.0 |
| 2 **SUPMER (only unlabeled)** | 87.3 | 46.7 | **84.0** | 89.3 | **79.6** | **80.2** | **72.4** | **57.3** | **61.7** | **54.8** | **71.3** | **70.5** | **71.3** |
| 3 PPT (labeled + unlabeled) | 84.7 | 45.0 | 82.4 | 87.8 | 67.2 | 77.4 | 64.3 | 55.3 | 61.6 | 53.9 | 68.9 | 67.7 | 68.0 |
| 4 MetaPT (labeled + unlabeled) | 86.1 | 46.3 | 83.7 | 89.4 | 73.8 | 80.1 | 67.2 | 57.4 | 60.0 | 54.3 | 70.1 | 69.9 | 69.9 |
| 5 **SUPMER (labeled + unlabeled)** | **89.1** | **48.2** | **85.7** | **90.8** | 79.3 | **83.4** | **73.2** | **58.8** | **63.7** | 55.3 | 70.5 | **71.5** | **72.5** |

Table 3: Results of few-shot learning on T5-base, considering different data and methods for prompt initialization.

actly the same unlabelled data as PPT and Unified-PPT for soft prompt initialization. Yet it considerably outperforms these two baselines, demonstrating that *the performance improvement is primarily attributable to our methodology rather than the meta-training data itself.* Additionally, SUPMER outperforms baseline methods with more tunable parameters (*e.g.*, full-model tuning) on the majority of datasets, achieving superior performance with fewer parameters.

Second, SUPMER is superior to all baselines in almost all domain-generalization setups. For example, compared to MetaPT which meta-trains soft prompts with a supervised sentiment analysis dataset, SUPMER exhibits average gains of 1.1% on T5-base and 1.4% on Flan-T5-XL. So it can be inferred that SUPMER shows stronger robustness to domain shifts, exhibiting better generalization to unseen tasks or domains.

Third, for both few-shot learning and domain generalization on Flan-T5-XL, SUPMER demonstrates superior performance across almost all datasets and domains in contrast to few-shot inference. It provides further evidence that for LMs such as Flan-T5-XL with inherent few-shot inference capabilities, our approach can significantly enhance their abilities in a parameter-efficient tuning strategy, without providing any in-context examples during inference.

Fourth, SUPMER also results in lower variances on most datasets. Few-shot learning is often notorious for its instability. And in our method we keep few-shot prompt tuning more stable.

### 4.4 Ablation Study

**Analysis of Generalization.** Figure 3 shows the performance trend for each method after different training steps on datasets CB and MRPC with T5-base model. It illustrates that few-shot prompt tuning converges slowly with its performance typically showing an overall decline during the final training steps because they may easily result in overfitting. In comparison, SUPMER achieves faster, stronger, and more enduring few-shot generalization. *It not*

| | Methods | Few-shot Learning | DG |
|---|---|---|---|
| 1 | only sp | 66.7 | 85.7 |
| 2 | only mc | 66.9 | 85.7 |
| 3 | only ss | 67.4 | 86.7 |
| 4 | w/o ta | 68.6 | 85.3 |
| 5 | w/o curriculum | 69.9 | 87.0 |
| 6 | w/o mgr | 69.4 | 86.1 |
| 7 | **SUPMER** | **71.3** | **88.0** |

Table 4: Results of ablation study to illustrate the effect of individual components. We report the average accuracy over all 12 datasets in few-shot learning and all 6 domains in domain generalization (DG).

*only accelerates the convergence to the optimal performance realizing fast adaptation, but also consistently maintains its optimal performance across prolonged training periods.*

**Effect of Sample Size.** We also discuss how the performance of SUPMER and other baselines varies when the number of training samples increases on SST-5 and SUBJ. As shown in Figure 4, with T5-base as the underlying PLM, when the number of training samples per label grows from 4 to 64, SUPMER is consistently better than other prompt tuning methods. And the performance gap between these methods is gradually reduced as the number of training data increases.

**Self-Supervised v.s. Supervised.** To illustrate that self-supervised meta-learning can better generalize to unseen tasks compared to supervised meta-learning, we also collect a set of labeled datasets (ensuring no overlap with downstream testing datasets) to formulate meta-training tasks for soft prompt initialization and conduct the experiments of few-shot learning on T5-base. The results are displayed in Table 3 (rows 1 and 2). As our collected labeled data contains lots of sentiment analysis datasets (*e.g.,* Yelp5), SUPMER (only labeled) and SUPMER (only unlabeled) reveal proximity in their performance on sentiment analysis tasks (*i.e.,* SST-2, SST-5, MR, CR). But in other tasks, using unlabeled data consistently achieves better results than utilizing only labeled data, also with a higher average accuracy over all datasets, which validates

the superiority of self-supervised meta-learning.

**Effect of integrating Labeled Data.** To further explore the impact of integrating labeled data and substantiate the efficacy of SUPMER following this integration, we amalgamate the original unlabeled meta-training data with our collected labeled data mentioned above, with a mixing ratio of labeled to unlabeled as 1:2. The amalgamated data is employed for constructing meta-training tasks to meta-train SUPMER. Moreover, following PPT (Gu et al., 2022) and MetaPT (Huang et al., 2022), We also leverage pre-training and vanilla MAML to initialize soft prompts using the same amalgamated data. The experimental results of few-shot learning on T5-base are shown in Table 3 (rows 3-5). First, we can see that SUPMER (labeled+unlabeled) outperforms SUPMER (unlabeled) and SUPMER (labeled) as it allows us to harness the high-quality advantages of labeled data while also exploiting the broader semantic concepts encapsulated by unlabeled data. Second, After the integration of labeled data, SUPMER still consistently demonstrates significantly superior performance compared to baseline methods employing the same data for prompt initialization, which further underscores the effectiveness of SUPMER.

**Effect of Individual Components.** We train the following ablation models. 1) only sp / mc / ss: we retain sentence-pair classification / multi-choice classification / single-sentence classification as the only anchor meta-training task format. 2) w/o ta: we entirely remove the task augmentation method. 3) w/o curriculum: we only retain the vanilla task augmentation without the curriculum-based idea. 4) w/o mgr: we remove the meta-gradient regularization function. All experiments follow the settings in §4.1 and are conducted on T5-base. We report the average accuracy of few-shot learning and domain generalization in Table 4. More detailed results are in Appendix D.

The results of Row 1-3 indicate that considering diversified task formats during meta-training helps efficiently generalize to different tasks as downstream tasks often contain various task formats. Row 4 and Row 5 highlight that task augmentation plays an essential role in our framework, with curriculum-based augmentation further enriching the task distribution and realistically simulating the distribution shift. Moreover, Row 6 validates the superiority of meta-gradient regularization in

avoiding overfitting to some domain-specific correlations, thus achieving better performance.

## 5    Conclusion

In this paper, we present SUPMER, a self-supervised meta-prompt learning framework with meta-gradient regularization. With a diverse set of well-designed self-supervised meta-training tasks, SUPMER jointly meta-learns a universal prompt initialization and an effective gradient regularization function for efficient few-shot generalization. Extensive experiments on few-shot learning and domain generalization show that SUPMER outperforms other prompt methods and full-model tuning, achieving state-of-the-art performance.

## Limitations

Although SUPMER performs superbly in a variety of problem scenarios, there still exist some limitations in our work: 1) We did not conduct any data filtering or cleaning operations to the meta-training data, which could potentially result in the inclusion of some biased content. 2) Our experiments are solely conducted on English tasks, and also do not involve some kinds of NLP tasks (*e.g.*, language generation Li et al., 2022c) or vision-language tasks (Zhang et al., 2022b; Li et al., 2022b; Zhang et al., 2019; Li et al., 2021).

To address these limitations, in the future we plan to conduct further cleansing and filtering on the current meta-training data. Besides, we intend to evaluate the few-shot performance of our framework in the multilingual setting and also broaden the scope of tasks, including retrieval (Pan et al., 2023), language generation (Li et al., 2022c) and vision-language tasks (Li et al., 2023b; Chen et al., 2023; Li et al., 2022a; Zhang et al., 2022a). Furthermore, we hope our work could pave the way for future research on better leveraging parameter-efficient methods under few-shot settings.

## Acknowledgements

This work has been supported in part by the Zhejiang NSF (LR21F020004), Key Research and Development Projects in Zhejiang Province (No. 2023C01030, 2023C01032), NSFC (No. 62272411), National Key Research and Development Program of China (2018AAA0101900), Ant Group and Alibaba-Zhejiang University Joint Research Institute of Frontier Technologies.

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

# Appendices

# A   Additional Information for SUPMER

## A.1   Complete Analysis of SUPMER

In this section, we provide a more comprehensive and complete analysis of SUPMER. We will show that during meta-training, the optimization of soft prompt embeddings $\theta$ and the meta-gradient regularization parameters $\phi$ tends to maximize the inner product of gradients obtained from the support set after regulation and gradients from the query set.

Specifically, to update the parameters $\theta$ and $\phi$, we should evaluate their gradients at first, denoting them as $g^\theta$ and $g^\phi$. Considering the original algorithm of MAML, each task consists of a support set and a query set. And only one step of gradient descent is applied in the inner-loop optimization. To make our statement more direct, we denote the loss function based on the support set and the query set as $\mathcal{L}_0$ and $\mathcal{L}_1$. In SUPMER, ignoring the regularized loss, only $\mathcal{L}_1$ is directly utilized to optimize $\phi$, while $\theta$ is optimized in a bi-level meta-optimization paradigm. Here we define the following terms related to $\theta$ similar to Nichol et al. (2018):

$$g_i^\theta = \frac{\partial \mathcal{L}_i(\theta_i)}{\partial \theta_i} \qquad \text{(gradient obtained during SGD)}$$

$$\overline{g}_i^\theta = \frac{\partial \mathcal{L}_i(\theta_0)}{\partial \theta_0} \qquad \text{(gradient at initial point)}$$

$$\overline{H}_i^\theta = \frac{\partial^2 \mathcal{L}_i(\theta_0)}{\partial \theta_0^2} \qquad \text{(Hessian at initial point)}$$

$$\theta_1 = \theta_0 - \alpha_1 \psi_\phi(g_0^\theta) \qquad \text{(gradient descent in the inner-loop)}$$

$$\tag{11}$$

For each definition $i \in \{0, 1\}$ and $\psi_\phi(\cdot)$ is the meta-gradient regularization operation. $\theta_0$ denotes the initial soft prompt embeddings for each step, and $\theta_1$ denotes the embeddings after the inner-loop optimization. Obviously we have $g_0^\theta = \overline{g}_0^\theta$. Firstly we perform a Taylor series expansion to approximate the SGD gradients $g_1^\theta$ obtained from the query set as follows:

$$
\begin{aligned}
g_1^\theta &= \frac{\partial \mathcal{L}_1(\theta_1)}{\partial \theta_1} \\
&= \frac{\partial \mathcal{L}_1(\theta_0)}{\partial \theta_0} + \frac{\partial^2 \mathcal{L}_1(\theta_0)}{\partial \theta_0^2}(\theta_1 - \theta_0) + \underbrace{O(||\theta_1 - \theta_0||^2)}_{=O(\alpha_1^2)} \\
&= \overline{g}_1^\theta - \alpha_1 \overline{H}_1^\theta \psi_\phi(\overline{g}_0^\theta) + O(\alpha_1^2)
\end{aligned}
$$

$$\tag{12}$$

Then we analysis the gradient descent operation in the inner-loop optimization based on the support set. Define $U$ as the gradient descent and we have $U(\theta_0) = \theta_0 - \alpha_1 \psi_\phi(\frac{\partial \mathcal{L}_0(\theta_0)}{\partial \theta_0})$. So we can get $\frac{\partial U(\theta_0)}{\partial \theta_0}$ and $\frac{\partial U(\theta_0)}{\partial \phi}$ as follows:

$$
\begin{aligned}
\frac{\partial U(\theta_0)}{\partial \theta_0} &= \frac{\partial}{\partial \theta_0}(\theta_0 - \alpha_1 \psi_\phi(\frac{\partial \mathcal{L}_0}{\partial \theta_0})) \\
&= I - \alpha_1 \frac{\partial \psi_\phi(g_0^\theta)}{\partial g_0^\theta} \cdot \frac{\partial g_0^\theta}{\partial \theta_0} \\
&= I - \alpha_1 \frac{\partial \psi_\phi(g_0^\theta)}{\partial g_0^\theta} \cdot \overline{H}_0^\theta
\end{aligned}
$$

$$\tag{13}$$

$$
\begin{aligned}
\frac{\partial U(\theta_0)}{\partial \phi} &= \frac{\partial}{\partial \phi}(\theta_0 - \alpha_1 \psi_\phi(\frac{\partial \mathcal{L}_0}{\partial \theta_0})) \\
&= -\alpha_1 \frac{\partial \psi_\phi(g_0^\theta)}{\partial \phi}
\end{aligned}
$$

$$\tag{14}$$

So based on Eq. (12, 13, 14), we can finally approximate the gradients $g^\theta$ and $g^\phi$ as:

$$
\begin{aligned}
g^\theta &= \frac{\partial \mathcal{L}_1(\theta_1)}{\partial \theta_0} \\
&= \frac{\partial \mathcal{L}_1(U(\theta_0))}{\partial \theta_0} \\
&= \frac{\partial \mathcal{L}_1}{\partial \theta_1} \cdot \frac{\partial U(\theta_0)}{\partial \theta_0} \\
&= \overline{g}_1^\theta - \alpha_1 \overline{H}_1^\theta \psi_\phi(\overline{g}_0^\theta) - \alpha_1 \overline{g}_1^\theta \cdot \frac{\partial \psi_\phi(\overline{g}_0^\theta)}{\partial \overline{g}_0^\theta} \cdot \overline{H}_0^\theta + O(\alpha_1^2) \\
&= \overline{g}_1^\theta - \alpha_1 \frac{\partial}{\partial \theta_0}(\overline{g}_1^\theta \psi_\phi(\overline{g}_0^\theta)) + O(\alpha_1^2)
\end{aligned}
$$

$$
\begin{aligned}
g^\phi &= \frac{\partial \mathcal{L}_1(\theta_1)}{\partial \phi} \\
&= \frac{\partial \mathcal{L}_1(U(\theta_0))}{\partial \phi} \\
&= \frac{\partial \mathcal{L}_1}{\partial \theta_1} \cdot \frac{\partial U(\theta_0)}{\partial \phi} \\
&= -\alpha_1 \overline{g}_1^\theta \frac{\partial \psi_\phi(\overline{g}_0^\theta)}{\partial \phi} + O(\alpha_1^2) \\
&= -\alpha_1 \frac{\partial}{\partial \phi}(\overline{g}_1^\theta \psi_\phi(\overline{g}_0^\theta)) + O(\alpha_1^2)
\end{aligned}
$$

$$\tag{15}$$

Thus, $-\frac{\partial}{\partial \theta_0}(\overline{g}_1^\theta \psi_\phi(\overline{g}_0^\theta))$ and $-\frac{\partial}{\partial \phi}(\overline{g}_1^\theta \psi_\phi(\overline{g}_0^\theta))$ indicate the optimization direction, which increases the inner product between gradients from the query set and gradients from the support set after transformation. To further consolidate our analysis, we also track the normalized gradient inner product in the first 5000 steps during meta-training. As shown in Figure 5, it is clear that the normalized gradient inner product gradually increases during meta-training.

On this basis, since there exists distribution shift between the support set and the query set after task augmentation, our method aligns the gradient directions across different distributions, which helps enhance model generalization. In other words, the trainable parameters descend in a coordinated manner such that the input-output correspondence is as

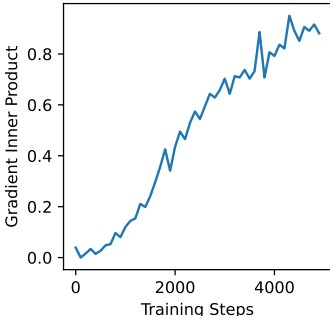

Figure 5: Normalized gradient inner products in the first 5000 steps during meta-training.

close as possible across two distributions with deviation. Besides, the meta-gradient regularization parameters $\phi$ also retain some domain-invariant information of the meta-training data in the above process. Considering that $\phi$ is fixed in downstream tasks, $\phi$ can be applied to encourage the alignment between the domain-specific gradients and avoid prompt-tuning overfitting to some domain-specific correlations.

## A.2 Constructing Anchor Meta Tasks

Given a sentence $x$ from unlabeled corpora, we can derive semantically meaningful sentence embedding $\boldsymbol{H} = f_\theta^{enc}(x)$ with PLMs, e.g., T5 encoder. And we apply K-means to cluster these unlabeled sentences according to their embeddings:

$$\mathcal{P}, \{\boldsymbol{\mu}_c\} = \arg\min_{\{\mathcal{C}_c\},\{\boldsymbol{\mu}_c\}} \sum_{c=1}^{K} \sum_{\boldsymbol{H} \in \mathcal{C}_c} \|\boldsymbol{H} - \boldsymbol{\mu}_c\|^2 \quad (16)$$

where $\boldsymbol{\mu}_c$ indicates the learned centroid of cluster $\mathcal{C}_c$ and $\mathcal{P}$ indicates the partitions of all sentences. K-means clustering leads to more abundant formats and objectives of meta-training tasks. Based on the results of K-means, we design three formats of anchor self-supervised meta-training tasks: sentence-pair classification, multi-choice classification, and single-text classification. Here we introduce each of them in detail.

**Sentence-pair Classification.** Sentence-pair classification takes a pair of sentences $(x_0, x_1)$ as input, and $x_0$ is the anchor sentence. We carry on next sentence prediction task and sentence similarity task in sentence-pair classification with the label list $\mathcal{Y} = [0, 1, 2]$. For the former one, following Gu et al. (2022), we set two sentences next to each other as label 0, those from the same document but not adjacent as label 2, and those from different documents as label 1. And for

---

**Algorithm 1** Meta-training Process of SUPMER

1: $p(\mathcal{T})$ : Distribution over anchor tasks
2: $f_\theta$ : PLM with soft prompt embeddings $\theta$
3: $\psi_\phi$ : Meta-gradient regularization
4: $\alpha_1, \beta_1, \beta_2$ : Learning rate
5: **TA** : Task augmentation in Algorithm 2

6: $s \leftarrow -1$
7: Randomly initialize $\theta, \phi$
8: **while** not done **do**
9:     Sample a batch of task $\{\tau_i\}_{i=1}^n$ from $p(\mathcal{T})$
10:     $\{\tau_i\}_{i=1}^n = \mathbf{TA}(\{\tau_i\}_{i=1}^n, p(\mathcal{T}), s)$
11:     **for all** $\tau_i = \{\mathcal{D}_{\tau_i}^s, \mathcal{D}_{\tau_i}^q\}$ **do**
12:         Evaluate $\nabla_\theta \mathcal{L}_{\mathcal{D}_{\tau_i}^s}(f_\theta)$ with $\mathcal{D}_{\tau_i}^s$
13:         Evaluate $\nabla_\theta \mathcal{L}_{\mathcal{D}_{\tau_i}^q}(f_\theta)$ with $\mathcal{D}_{\tau_i}^q$
14:         Transform $\nabla_\theta \mathcal{L}_{\mathcal{D}_{\tau_i}^s}(f_\theta)$ via $\psi_\phi(\cdot)$
15:         $s_i = \dfrac{\nabla_\theta \mathcal{L}_{\mathcal{D}_{\tau_i}^q}(f_\theta) \cdot \psi_\phi(\nabla_\theta \mathcal{L}_{\mathcal{D}_{\tau_i}^s}(f_\theta))}{\|\nabla_\theta \mathcal{L}_{\mathcal{D}_{\tau_i}^q}(f_\theta)\| \cdot \|\psi_\phi(\nabla_\theta \mathcal{L}_{\mathcal{D}_{\tau_i}^s}(f_\theta))\|}$
16:         $\theta_i' = \theta - \alpha_1 \psi_\phi(\nabla_\theta \mathcal{L}_{\mathcal{D}_{\tau_i}^s}(f_\theta))$
17:     **end for**
18:     $s \leftarrow \sum_i s_i / \sum_i 1$
19:     $\theta \leftarrow \theta - \beta_1 \nabla_\theta \sum_{\tau_i} \mathcal{L}_{\mathcal{D}_{\tau_i}^q}(f_{\theta_i'})$
20:     $\phi \leftarrow \phi - \beta_2 \nabla_\phi \big( \sum_{\tau_i} \mathcal{L}_{\mathcal{D}_{\tau_i}^q}(f_{\theta_i'}) + \mathcal{L}_{reg} \big)$
21: **end while**
22: **return** $\theta, \phi$

---

sentence similarity task, we set two sentences coming from the same cluster as label 0, and those from different clusters as label 1. In this way, the prompt template and verbalizer are designed as:

$$\begin{aligned} P &= \text{``}s_1 \langle \text{X} \rangle .s_2\text{''} \\ \mathcal{V} &= \{0 \to \text{yes}, 1 \to \text{no}, 2 \to \text{maybe}\} \end{aligned} \quad (17)$$

**Multi-choice Classification.** Multi-choice classification takes an anchor sentence $x_0$ as the query and we should find the correct one in several answer candidates. Here we also set two different tasks. The first one aims to select the sentence next to $s_0$ and the second one aims to select the sentence which belongs to the same cluster as $s_0$. In each task we will set four candidates, and only one of them is correct. We design the prompt template and verbalizer as follows:

$$\begin{aligned} P &= \text{``}s_0? \text{ A.}s_1 \cdots \text{D.}s_4. \text{ Answer: } \langle \text{X} \rangle \text{''} \\ \mathcal{V} &= \{0 \to \text{A}, 1 \to \text{B}, 2 \to \text{C}, 3 \to \text{D}\} \end{aligned} \quad (18)$$

**Single-Sentence Classification.** Through K-means clustering, each sentence is associated with a cluster label $r_i$ in $\{0, 1\}^K$ where $r_{ic} = 1$ if $c = k$ and $y_{ic} = 0$ if $c \neq k$. Here $k$ represents the cluster

to which the sentence belongs. We simply use $r_i$ as the pseudo label for meta-training and construct 4-way classification tasks. As for the designing of the verbalizer, we transform the single-sentence classification into the format of multi-choice classification. We insert the centroid of cluster $\boldsymbol{\mu}_c$ into the template and use it to represent the corresponding cluster. So that we have:

$$P = \text{``}s_0?\text{ A. }\langle \mu_{c_1} \rangle \cdots \text{D. }\langle \mu_{c_4} \rangle \text{ . Answer: }\langle X \rangle \text{''}$$
$$\mathcal{V} = \{0 \rightarrow \text{A}, 1 \rightarrow \text{B}, 2 \rightarrow \text{C}, 3 \rightarrow \text{D}\} \quad (19)$$

On this basis, for each task format, we separate all data into different tasks to construct anchor meta-training tasks with good task distributions. Through K-means, sentences with similar embeddings are clustered into the same group. So in sentence-pair classification and multi-choice classification, we group samples whose anchor sentence comes from the same cluster into the same meta-training task. And in single-sentence classification, for each meta-training task, we randomly select $N$ clusters as $N$ classes and then sample $k$ sentences for each cluster to construct a $N$-way $k$-shot classification task ($N = 4$). In this way, we completely construct all anchor meta-training tasks.

### A.3 Additional Loss to Train Meta-Gradient Regularization Parameters

In the meta-training stage, we optimize the meta-gradient regularization parameters $\phi$ via Eq. (7), utilizing the same loss which optimizes the soft prompt embeddings. Here we introduce a regularized loss to attach some additional restrictions when updating the meta-gradient regularization parameters. Notably, a higher value of $b_k$ in Eq. (10) indicates a higher probability of a larger distribution deviation between the support set and the query set. Furthermore, in Eq. (5) we also tend to increase $z$ to achieve a more pronounced gradient transformation with a more noticeable distribution deviation. From this perspective, $z$ has a similar monotonicity with $b_k$, and they both range between 0 and 1. Thus we further add a regularized loss $\mathcal{L}_{reg} = \|z - b_k\|^2$ to constrain the value of $z$ and finally modify Eq. (7) into:

$$\phi \leftarrow \phi - \beta_2 \nabla_\phi \Big( \sum_{\tau_i \sim p(\mathcal{T})} \mathcal{L}_{\mathcal{D}_{\tau_i}^q}(f_{\theta_i'}) + \lambda \mathcal{L}_{reg} \Big) \quad (20)$$

### A.4 Pseudo-Codes of SUPMER

We show the pseudo-codes for the meta-training process of SUPMER in Alg. 1. And the process of curriculum-based task augmentation is described in Alg. 2.

---

**Algorithm 2 TA : Curriculum-based Task Augmentation**

1: $\{\tau_i\}_{i=1}^n$ : A batch of anchor tasks
2: $p(\mathcal{T})$ : Distribution over anchor tasks
3: $s \in [-1, 1]$ : Avg cos-sim between gradients
4: $\alpha, m$ : hyper-parameters

5: $s \leftarrow (1 + s)/2$
6: $b \leftarrow (m^s - 1)/(m - 1)$
7: **for all** $\tau_i = \{\mathcal{D}_{\tau_i}^s, \mathcal{D}_{\tau_i}^q\}$ **do**
8:      Sample task $\tau_j = \{\mathcal{D}_{\tau_j}^s, \mathcal{D}_{\tau_j}^q\}$ from $p(\mathcal{T})$
9:      Draw $\lambda$ from $Beta(\alpha, b\alpha)$
     // $\mathcal{D}_{\tau_i}^q = (\boldsymbol{H}_i^q, \boldsymbol{Y}_i^q), \mathcal{D}_{\tau_j}^q = (\boldsymbol{H}_j^q, \boldsymbol{Y}_j^q)$
     // $\boldsymbol{H}$: the hidden representations of samples
10:      $\tilde{\boldsymbol{H}}_i^q = (1 - \lambda)\boldsymbol{H}_i^q + \lambda \boldsymbol{H}_j^q$
11:      $\tilde{\boldsymbol{Y}}_i^q = (1 - \lambda)\boldsymbol{Y}_i^q + \lambda \boldsymbol{Y}_j^q$
12:      $\mathcal{D}_{\tau_i}^q \leftarrow (\tilde{\boldsymbol{H}}_i^q, \tilde{\boldsymbol{Y}}_i^q)$
13: **end for**
14: **return** $\{\tau_i\}_{i=1}^n$

---

## B Dataset & Baseline Details

**Few-shot Learning.** We conduct experiments of few-shot learning on 6 different downstream English tasks with 12 datasets. Since some of the test sets of the datasets are not publicly available, following Karimi Mahabadi et al. (2022), we leverage the original validation sets of SST-2, CB, RTE, QNLI, WiC, MRPC, and QQP[1] as substitutes for the unavailable test sets. And the validation sets for few-shot learning are sampled from the original training set, ensuring no overlap with our designated test sets. Besides, we download the datasets of SST-2, SST-5, MR, CR, and SUBJ from Gao et al. (2021). And the rest of the datasets are obtained from the HuggingFace Datasets library (Lhoest et al., 2021). CB, RTE, BoolQ, and Wic are from SuperGLUE Benchmark (Wang et al., 2019), while QNLI, MRPC, and QQP are from GLUE Benchmark (Wang et al., 2018) with Creative Commons license (CC BY 4.0). We give the statistics of all these datasets in Table 5.

**Domain Generalization.** Similar to Calderon et al. (2022), We evaluate on the sentiment analysis task including 6 different domains: Airlines (A), Books (B), DVDs (D), Electronics (E), Kitchen appliances (K), and Restaurants (R). Each domain has totally 2,000 manually labeled data of binary categories for testing, including 1000 positive and

---

[1] https://quoradata.quora.com/

| Dataset | Task | #Train | #Test | K |
|---------|------|--------|-------|---|
| SST-2 | Sentiment analysis | 6920 | 872 | 2 |
| SST-5 | Sentiment analysis | 8544 | 2210 | 5 |
| MR | Sentiment analysis | 8662 | 2000 | 2 |
| CR | Sentiment analysis | 1774 | 2000 | 2 |
| SUBJ | Subjectivity classification | 8000 | 2000 | 2 |
| TREC | Question classification | 5452 | 500 | 6 |
| CB | Natural language inference | 250 | 56 | 3 |
| RTE | Natural language inference | 2490 | 277 | 2 |
| QNLI | Question answering | 104743 | 5463 | 2 |
| WiC | Word sense disambiguation | 5428 | 638 | 2 |
| MRPC | Paraphrase detection | 3668 | 408 | 2 |
| QQP | Paraphrase detection | 363846 | 40430 | 2 |

Table 5: Statistics of all 12 datasets for few-shot learning. K is the number of labels. We sample $N \times K$ instances from the original training set to construct the few-shot training and validation sets. And *#Test* shows the size of the test set.

1000 negative. We choose A as the source domain and the other five (B, D, E, K, R) constitute the target domains. We sample 16 instances per label from the training set of the source domain for prompt tuning and then evaluate on the test sets of all 6 domains.

**Baselines.** We first compare with baseline methods with the same number of parameters as SUPMER. These methods utilize prompt tuning (Lester et al., 2021) to handle downstream tasks, with the key distinction lying in the initialization of the soft prompts. Vallina prompt tuning (**PT** Lester et al., 2021) directly tunes the soft prompts in the downstream task, which are randomly initialized from a normal distribution. **PPT** (Gu et al., 2022) pretrains soft prompts in a self-supervised way with 3 formats of pre-training tasks: sentence-pair classification, multiple-choice classification and single-text classification. **Unified-PPT** (Gu et al., 2022) formulate all these three formats into a unified task form. **MetaPT** (Huang et al., 2022) using a supervised sentiment analysis dataset Yelp5 as the meta-training data and directly leveraging MAML to initialize soft prompts.

To further demonstrate the effectiveness of our method, we also consider baseline methods with more tunable parameters, including **Prefix-Tuning** (Li and Liang, 2021) and **P-tuning-v2** (Liu et al., 2022), which add prompts at each layer of PLM. We also compare with full-model tuning (**FT**) that fine-tunes all parameters of the PLM.

Given that FLAN-T5-XL was also designed with few-shot inference in mind, we newly compare with two baseline methods on FLAN-T5-XL: **zero-**

| Hyper-parameter | Value |
|-----------------|-------|
| Number of clusters for each task format | 250 |
| Tasks per batch | 4 |
| Size of support set per task | 32 |
| Size of query set per task | 32 |
| Optimizer | Adam |
| Inner loop learning rate | 0.1 |
| Outer loop learning rate | 0.1 |
| Learning rate for $\phi$ | 1e-4 |
| Scheduler | Linear scheduler |
| Warm-up steps | 0 |
| Max training steps | 100,000 |
| Validation steps | 2,000 |
| Max sequence length | 512 |
| $\lambda$ | 1.0 |
| $m$ | 2.0 |

Table 6: Hyper-parameters for SUPMER. $\phi$ denotes the meta-gradient regularization parameters. $\lambda$ is the coefficient of the regularized loss. And $m$ is the curve parameter in the curriculum-based task augmentation.

**shot inference** and **few-shot inference**. For both of them, we directly employ Flan-T5-XL for downstream evaluation, coupled with carefully designed task instructions for each dataset. Furthermore, in few-shot inference, we also provide an appropriate number of few-shot examples to form a demonstration context.

## C Training Details

We apply the T5 base model (Raffel et al., 2020) (220M parameters) and Flan-T5-XL model (Chung et al., 2022) (3B parameters) as the underlying PLM, and use the HuggingFace Pytorch implementation (Wolf et al., 2020). We run experiments with 8 GeForce RTX 3090 24G GPUs. And the meta-training process of SUPMER takes about 140 GPU hours. Next we will describe the details of training hyper-parameters in the case of leveraging T5-base as the PLM.

### C.1 Training Hyper-parameters for Downstream Tasks

In our experiments, we leverage full-model tuning and prompt tuning to solve downstream tasks, including few-shot learning and domain generalization. In few-shot learning, following some prior work (Schick and Schütze, 2021; Karimi Mahabadi et al., 2022), we set the maximum sequence length of each example to 256 for CR, SUBJ, CB, RTE and WiC, and 128 for other datasets. While in domain generalization, the maximum sequence length of each example is set to 256.

We run each experiment 5 times on the random

| Methods | SST-2 | SST-5 | MR | CR | SUBJ | TREC | CB | RTE | QNLI | WiC | MRPC | QQP |
|---|---|---|---|---|---|---|---|---|---|---|---|---|
| only sp | $83.6_{1.5}$ | $42.6_{2.2}$ | $81.7_{1.8}$ | $86.0_{0.6}$ | $65.8_{2.8}$ | $71.2_{6.6}$ | $64.6_{2.1}$ | $57.0_{2.5}$ | $58.4_{3.3}$ | $53.6_{1.5}$ | $69.9_{1.3}$ | $66.0_{1.0}$ |
| only mc | $83.4_{1.4}$ | $44.5_{1.9}$ | $79.3_{5.1}$ | $88.3_{0.5}$ | $70.5_{4.7}$ | $66.4_{1.4}$ | $65.9_{3.1}$ | $54.9_{1.3}$ | $58.7_{1.7}$ | $54.2_{1.8}$ | $68.8_{0.8}$ | $67.6_{1.3}$ |
| only ss | $84.5_{1.5}$ | $45.0_{2.0}$ | $81.5_{0.7}$ | $88.4_{0.5}$ | $73.3_{3.1}$ | $79.1_{1.4}$ | $62.1_{2.6}$ | $53.9_{1.0}$ | $56.5_{1.4}$ | $53.3_{1.3}$ | $67.7_{1.3}$ | $63.7_{1.7}$ |
| w/o ta | $84.7_{1.0}$ | $40.1_{3.3}$ | $81.9_{1.8}$ | $87.2_{0.8}$ | $73.6_{2.8}$ | $78.8_{3.7}$ | $66.4_{1.9}$ | $56.6_{0.9}$ | $59.4_{1.8}$ | $54.3_{2.4}$ | $69.5_{1.1}$ | $70.2_{1.1}$ |
| w/o curriculum | $86.8_{0.8}$ | $40.8_{2.2}$ | $82.3_{1.3}$ | $88.4_{0.9}$ | $74.8_{3.1}$ | $79.7_{1.6}$ | $71.0_{2.1}$ | $56.5_{0.8}$ | $\mathbf{62.6_{1.4}}$ | $\mathbf{55.4_{1.1}}$ | $69.7_{0.8}$ | $\mathbf{71.3_{1.2}}$ |
| w/o mgr | $85.0_{1.3}$ | $44.5_{1.1}$ | $82.8_{0.7}$ | $88.0_{0.5}$ | $76.0_{1.7}$ | $79.5_{5.0}$ | $67.1_{1.6}$ | $56.8_{0.8}$ | $58.9_{2.4}$ | $54.4_{2.0}$ | $70.0_{1.0}$ | $70.3_{0.9}$ |
| **SUPMER** | $\mathbf{87.3_{0.5}}$ | $\mathbf{46.7_{0.6}}$ | $\mathbf{84.0_{0.6}}$ | $\mathbf{89.3_{0.3}}$ | $\mathbf{79.6_{2.2}}$ | $\mathbf{80.2_{0.9}}$ | $\mathbf{72.4_{1.4}}$ | $\mathbf{57.3_{1.0}}$ | $61.7_{1.0}$ | $54.8_{1.2}$ | $\mathbf{71.3_{0.5}}$ | $70.5_{1.0}$ |

Table 7: Detailed results of ablation study for few-shot learning to illustrate the effect of individual Components. In the first three rows we keep only one anchor task format during meta-training, and sp stands for sentence-pair classification, mc for multi-choice classification, ss for single-sentence classification. And w/o ta means entirely removing task augmentation, w/o curriculum only retains the vanilla task augmentation without the curriculum-based idea. w/o mgr means removing the meta-gradient regularization method.

| Method | Source | Target | | | | | |
|---|---|---|---|---|---|---|---|
| | **A** | **B** | **D** | **E** | **K** | **R** |
| only sp | $83.4_{1.1}$ | $82.1_{1.4}$ | $83.0_{0.7}$ | $88.5_{1.2}$ | $88.9_{0.8}$ | $88.1_{0.7}$ |
| only mc | $84.0_{0.6}$ | $82.3_{1.2}$ | $81.5_{0.9}$ | $88.5_{1.0}$ | $89.3_{0.7}$ | $88.8_{0.7}$ |
| only ss | $83.6_{0.7}$ | $84.7_{0.8}$ | $84.2_{0.6}$ | $88.9_{0.9}$ | $89.7_{0.9}$ | $89.0_{0.3}$ |
| w/o ta | $83.4_{0.8}$ | $82.0_{1.4}$ | $81.7_{1.5}$ | $87.8_{0.8}$ | $88.2_{0.6}$ | $88.6_{0.6}$ |
| w/o curriculum | $84.0_{0.5}$ | $84.7_{0.8}$ | $83.9_{0.6}$ | $89.6_{0.5}$ | $90.3_{0.8}$ | $89.7_{1.1}$ |
| w/o mgr | $83.8_{0.4}$ | $83.4_{0.5}$ | $83.3_{0.5}$ | $88.1_{0.6}$ | $89.2_{0.8}$ | $88.9_{0.4}$ |
| **SUPMER** | $\mathbf{85.7_{0.5}}$ | $\mathbf{85.3_{0.6}}$ | $\mathbf{85.1_{0.4}}$ | $\mathbf{90.3_{0.7}}$ | $\mathbf{91.1_{0.5}}$ | $\mathbf{90.4_{0.4}}$ |

Table 8: Detailed results of ablation study for domain generalization to illustrate the effect of individual Components.

seed [10, 20, 30, 40, 50] and report the average accuracy as well as the standard deviation. For both full-model tuning and prompt tuning, We implement AdamW as the optimizer. We use a batch size of 32 and train the model for 200 epochs, meanwhile evaluating the model every 10 steps. And we report the results for hyper-parameters performing the best on the validation set for each task.

Besides, for full-model tuning, all parameters of PLM are fine-tuned without adding soft prompts. We use the learning rate of [1e-5, 2e-5, 3e-5] and choose the one obtaining the highest validation performance. Moreover, to fine-tune the Flan-T5-XL model, we use ZeRO (Rajbhandari et al., 2020) stage-2 provided in DeepSpeed (Rasley et al., 2020) to reduce GPU memory usage.

For prompt tuning, we freeze all PLM parameters and only tune soft prompts composed of 100 soft tokens. As a result, the tunable parameters of prompt tuning are only 77K with T5-base and 205K with Flan-T5-XL, updating around 3000 and 15000 times fewer parameters on T5-base and Flan-T5-Xl, respectively, compared to full-model tuning. And we find that prompt tuning requires a much larger learning rate than full-model tuning. We search for the learning rate in [1e-1, 2e-1, 3e-1] and also choose the model with the best performance on the

validation set.

## C.2 Training Hyper-parameters for Prompt Initialization

**Pre-training for prompt initialization.** Gu et al. (2022) proposes two frameworks for unsupervised prompt pre-training, named PPT and Unified PPT. PPT designs three formats of unsupervised pre-training tasks (sentence-pair classification, multiple-choice classification and single-text classification), and Unified-PPT further formulate them into a unified task form. We implement PPT and Unified-PPT following the hyper-parameters provided in Gu et al. (2022) and reset the pre-trained language model to T5-base and Flan-T5-XL. Specifically, for both PPT and Unified-PPT, we sample 10GB of unlabeled data from OpenWeb-Text to construct pre-training tasks for each task format. And 5% data are split for validation. We apply the "inverse square root" learning rate scheduler with no warm-up steps and set the learning rate as 0.1. We set the batch size to 256 with the max sequence length as 512, and train soft prompts for at most 200,000 steps. We evaluate the performance on the validation set every 2,000 steps and choose prompts with the lowest validation loss.

| Method | Source | Target | | | | | AVG |
|---|---|---|---|---|---|---|---|
| | **A** | **B** | **D** | **E** | **K** | | |
| 1 SUPMER (only labeled) | **86.4** | 84.7 | 84.8 | 90.0 | 90.7 | | 87.3 |
| 2 SUPMER (only unlabeled) | 85.7 | **85.3** | **85.1** | **90.3** | **91.1** | | **87.5** |
| 3 PPT (labeled + unlabeled) | 83.1 | 79.0 | 84.4 | 89.3 | 90.6 | | 85.3 |
| 4 MetaPT (labeled + unlabeled) | 86.3 | 85.3 | 86.7 | 90.1 | 91.4 | | 88.0 |
| 5 SUPMER (labeled + unlabeled) | **86.6** | **88.6** | **88.5** | **92.7** | **93.7** | | **90.0** |

Table 9: Results of domain generalization on T5-base, considering different data and methods for prompt initialization. As our collected labeled data includes Yelp5, a sentiment analysis dataset in the domain of restaurants, we conduct the experiments of domain generalization only across domains A, B, D, E, K (without R).

**Meta-training for prompt initialization.** In our SUPMER framework, we sample 10GB of unlabeled data from OpenWebText to construct self-supervised meta-training tasks. We split 5% data to construct tasks for validation. And for each task format, we first set the number of clusters to 250. We sample 4 meta-training tasks in a batch, and train the prompt embeddings $\theta$ and the meta-gradient regularization parameters $\phi$ for at most 100,000 steps. We also evaluate the performance on the validation set every 2,000 steps, choosing $\theta$ and $\phi$ with the lowest validation loss for downstream tasks. Table 6 lists all training hyper-parameters for SUPMER. It is worth noting that for most hyper-parameters in Table 6, we just set a default value by experience without tuning them. we tune the hyper-parameters which are also tuned in other baselines (*e.g.*, learning rate), ensuring all methods have the same number of tunable hyper-parameters in our experiments.

Moreover, to illustrate the superiority of self-supervised meta-learning, we also imitate MetaPT(Huang et al., 2022) to initialize soft prompts via supervised meta-learning. MetaPT uses a supervised sentiment analysis dataset Yelp5 as the meta-training data, which has 650,000 training samples only covering the domain of restaurants. Following Huang et al. (2022), We group all labeled data into 10 clusters through K-means. And we set the inner loop learning rate to 0.08, the outer loop learning rate to 0.025 with the early stop patience as 6. Other hyper-parameters are consistent with those in SUPMER.

## D Full Results of Ablation Study

In this section, we first give detailed experimental results of the ablation study to illustrate the effect of individual components. We evaluate each ablation model over all 12 datasets of few-shot learning and all 6 domains of domain generalization, with

T5-base as the underlying PLM. We run each experiment 5 times on the random seed [10, 20, 30, 40, 50] and report the average performances as well as the standard deviation. The detailed results of few-shot learning and domain generalization are shown in Table 7 and Table 8. We can see each component is critical in our framework.

Besides, in §4.4, to explore the superiority of self-supervised meta-learning and the impact of integrating additional labeled data for soft prompt initialization, we conduct experiments of few-shot learning on T5-base, considering different data and methods for soft prompt initialization. We also carry out experiments of domain generation leveraging different data with various prompt initialization methods, with the results presented in Table 9. From Table 3 and 9, it is evident that self-supervised meta-learning utilizing unlabeled data exhibits enhanced adaptability to unseen tasks in comparison to its supervised counterparts. And amalgamating both labeled and unlabeled data for the construction of meta-training tasks emerges as a more advantageous strategy. When it comes to employing both labeled and unlabeled data for prompt initialization, SUPMER continues to showcase markedly superior results in contrast to baseline methods in the realms of both few-shot learning and domain generalization.