# OpenReview forum: "Self-supervised Meta-Prompt Learning with Meta-Gradient Regularization for Few-shot Generalization"
_EMNLP/2023/Conference — EMNLP 2023 Findings_

### Official Review · Reviewer_gNne · 2023-08-03

**Soundness:** 4

**Excitement:**

4: Strong: This paper deepens the understanding of some phenomenon or lowers the barriers to an existing research direction.

**Paper Topic And Main Contributions:**

This paper proposes a method for better soft prompt initialization, SUPMER.  SUPMER leverages self-supervised meta-learning with a diverse set of well-designed meta-training tasks to learn a universal prompt initialization for efficient adaptation using only unlabeled data. Extensive experiments show that SUPMER achieves better performance for different few-shot downstream tasks, and also exhibits a stronger domain generalization ability.

**Reasons To Accept:**

1. Intuitive method
2. Solid experiments
3. Thorough ablation study that answered a lot of potential questions.

**Reasons To Reject:**

I’m just a bit concerned about the setting of using only unlabeled data. I feel it is not a very real-world setting because we’ve already have a lot of NLP tasks with labelled data. Why not using them? If the authors can employ such labelled data using the proposed method, compare with proper baselines and still achieve superior performance, I will update my rating.

**Reproducibility:**

4: Could mostly reproduce the results, but there may be some variation because of sample variance or minor variations in their interpretation of the protocol or method.

**Reviewer Confidence:**

4: Quite sure. I tried to check the important points carefully. It's unlikely, though conceivable, that I missed something that should affect my ratings.

---

> ### Author Rebuttal · Authors · 2023-08-29
>
> We sincerely appreciate you for the constructive and insightful feedback! We are greatly encouraged by the recognition of our work's **intuitive methodology and solid experiments**. To better address the concern of employing labeled data, we have added additional parts of experiments to show (1) the impact of incorporating labeled data, and (2) the effectiveness of SUPMER compared with other labeled-data-integrated prompt initialization methods after the integration of labeled data.
>
> &nbsp;
>
> ### **1. The impact of incorporating labeled data**
>
> In order to explore whether the utilization of labeled data during the meta-training process could enhance the performance of SUPMER, we collect a set of labeled datasets to formulate meta-training tasks for soft prompt initialization. And then we conduct the experiments under three settings:
> (1) **SUPMER *(only labeled)*** utilizes only labeled data for constructing meta-training tasks.
> (2) **SUPMER *(only unlabeled)*** utilizes only unlabeled data for constructing meta-training tasks.
> (3) **SUPMER *(labeled + unlabeled)*** involves a combination of both labeled and unlabeled data for the construction of meta-training tasks, with a mixing ratio of labeled to unlabeled as 1:2.
>
> To improve the efficiency of our experiments, we employ the T5-base backbone for few-shot learning and domain generalization. From the results in the following tables, we have the following observations:
>
> (1) By comparing *SUPMER (only labeled)* and *SUPMER (only unlabeled)*, we can see that **self-supervised meta-learning with unlabeled data can better generalize to unseen tasks when compared to supervised meta-learning**. This can be attributed to the fact that unlabeled data generally covers broader semantic concepts. Specifically, as our labeled data contains some sentiment analysis datasets, *SUPMER (only labeled)* and *SUPMER (only unlabeled)* reveal proximity in their performance on sentiment analysis tasks (i.e., SST-2, SST-5, MR, CR). But in other tasks, using unlabeled data consistently achieves better results than utilizing only labeled data, also with a higher average accuracy over all datasets/domains.
>
> (2) *SUPMER (labeled+unlabeled)* also significantly outperforms *SUPMER (unlabeled)* and *SUPMER (labeled)* *(**72.5** v.s. 71.3 v.s. 70.0 in few-shot learning *&* **90.0** v.s. 87.5 v.s. 87.3 in domain generalization)*. Thank you for your constructive advice! The results further validate that **combining both labeled and unlabeled data for meta-training task construction is indeed a superior choice.** It allows us to harness the high-quality advantages of labeled data while also exploiting the broader semantic concepts encapsulated by unlabeled data. As a result, the utilization of both labeled and unlabeled data outperforms the use of either labeled or unlabeled data alone.
>
> &nbsp;
>
> **Experimental results of few-shot learning on T5-base**:
> | **Method & Dataset** | SST-2 | SST-5 | MR | CR | SUBJ | TREC | CB | RTE | QNLI | WiC | MRPC | QQP | *AVG* |
> | :--- | :---: | :---: | :---: | :---: | :---: | :---: | :---: | :---: | :---: | :---: | :---: | :---: | :---: |
> | SUPMER *(only labeled)* | 87.5 | 47.0 | 83.8 | 89.9 | 75.4 | 79.6 | 67.9 | 56.6 | 59.0 | 54.6 | 69.2 | 69.5 | 70.0 |
> | SUPMER *(only unlabeled)* | 87.3 | 46.7 | 84.0 | 89.3 | **79.6** | 80.2 | 72.4 | 57.3 | 61.7 | 54.8 | **71.3** | 70.5 | 71.3 |
> | SUPMER *(labeled + unlabeled)* | **89.1** | **48.2** | **85.7** | **90.8** | 79.3 | **83.4** | **73.2** | **58.8** | **63.7** | **55.3** | 70.5 | **71.5** | **72.5** |
>
> **Experimental results of domain generalization on T5-base**:
> | **Method & Domain** | B | D | E | K | A | *AVG* |
> | :--- | :---: | :---: | :---: | :---: | :---: | :---: |
> | SUPMER *(only labeled)* | 86.4 | 84.7 | 84.8 | 90.0 | 90.7 | 87.3 |
> | SUPMER *(only unlabeled)* | 85.7 | 85.3 | 85.1 | 90.3 | 91.1 | 87.5 |
> | SUPMER *(labeled + unlabeled)* | **86.6** | **88.6** | **88.5** | **92.7** | **93.7** | **90.0** |
>
> &nbsp;
>
> ### **2. The effectiveness of SUPMER compared with other labeled-data-integrated prompt initialization methods**
>
> ***To further prove SUPMER's efficacy after the integration of labeled data***, We compare SUPMER with other labeled-data-integrated prompt initialization methods. We apply ***the same combined dataset of labeled and unlabeled data to other baseline methods for prompt initialization***. Following PPT [A] and MetaPT [B], we leverage pre-training and vanilla MAML to initialize soft prompts using the combined dataset of labeled and unlabeled data. We denote these baselines as **PPT *(labeled + unlabeled)*** and **MetaPT *(labeled + unlabeled)***, respectively.
>
> The experimental results are listed in the tables below, from which we can draw the following conclusions. **First**, when combined with the results in Tables 1 and 2 of the main paper, it becomes evident that the utilization of both labeled and unlabeled data for initializing soft prompts leads to improved performance across all baseline methods that focus on prompt initialization.
>
> **Second, after the integration of labeled data, SUPMER still consistently demonstrates significantly superior performance** compared to baseline models that employ the same dataset for prompt initialization. Specifically, SUPMER exhibits an average performance improvement of 4.5 points over PPT and 2.6 points over MetaPT in few-shot learning. In terms of domain generalization, SUPMER elevates the average performance by 4.7 points compared to PPT and by 2.0 points compared to MetaPT. These results further underscore the effectiveness of SUPMER.
>
> &nbsp;
>
> **Experimental results of few-shot learning on T5-base**:
> | **Method & Dataset** | SST-2 | SST-5 | MR | CR | SUBJ | TREC | CB | RTE | QNLI | WiC | MRPC | QQP | *AVG* |
> | :--- | :---: | :---: | :---: | :---: | :---: | :---: | :---: | :---: | :---: | :---: | :---: | :---: | :---: |
> | FT | 83.6 | 41.2 | 81.7 | 88.3 | **80.0** | 79.8 | 71.9 | 56.9 | 62.3 | 54.6 | 70.2 | 69.5 | 70.0 |
> | PT | 71.9 | 37.3 | 73.2 | 84.4 | 61.5 | 65.3 | 58.9 | 53.2 | 55.2 | 53.1 | 66.6 | 63.0 | 62.0 |
> | PPT *(labeled + unlabeled)* | 84.7 | 45.0 | 82.4 | 87.8 | 67.2 | 77.4 | 64.3 | 55.3 | 61.6 | 53.9 | 68.9 | 67.7 | 68.0 |
> | MetaPT *(labeled + unlabeled)* | 86.1 | 46.3 | 83.7 | 89.4 | 73.8 | 80.1 | 67.2 | 57.4 | 60.0 | 54.3 | 70.1 | 69.9 | 69.9 |
> | SUPMER *(labeled + unlabeled)* | **89.1** | **48.2** | **85.7** | **90.8** | 79.3 | **83.4** | **73.2** | **58.8** | **63.7** | **55.3** | **70.5** | **71.5** | **72.5** |
>
> **Experimental results of domain generalization on T5-base**:
> | **Method & Domain** | B | D | E | K | A | *AVG* |
> | :--- | :---: | :---: | :---: | :---: | :---: | :---: |
> | FT | 84.4 | 83.9 | 81.0 | 84.1 | 85.0 | 83.7 |
> | PT | 79.8 | 75.3 | 76.0 | 79.6 | 79.8 | 78.1 |
> | PPT *(labeled + unlabeled)* | 83.1 | 79.0 | 84.4 | 89.3 | 90.6 | 85.3 |
> | MetaPT *(labeled + unlabeled)* | 86.3 | 85.3 | 86.7 | 90.1 | 91.4 | 88.0 |
> | SUPMER *(labeled + unlabeled)* | **86.6** | **88.6** | **88.5** | **92.7** | **93.7** | **90.0** |
>
> &nbsp;
>
> ***Note**: As our collected labeled data includes Yelp5, a sentiment analysis dataset in the domain of restaurants, we conduct the experiments of domain generalization only across domains B, D, E, K, A (without R). And then there is no overlap between our collected labeled data and the downstream testing datasets.*
>
> Overall, we sincerely thank you once again for your valuable suggestions! We will incorporate the aforementioned experiments in the next version.
>
>
> [A] Gu Y, Han X, Liu Z, et al. Ppt: Pre-trained prompt tuning for few-shot learning[J]. arXiv preprint arXiv:2109.04332, 2021.
>
> [B] Huang Y, Qian K, Yu Z. Learning a better initialization for soft prompts via meta-learning[J]. arXiv preprint arXiv:2205.12471, 2022.

---

### Official Review · Reviewer_XEMq · 2023-08-04

**Typos Grammar Style And Presentation Improvements:** line 168 -> space missing
**Soundness:** 3

**Excitement:**

4: Strong: This paper deepens the understanding of some phenomenon or lowers the barriers to an existing research direction.

**Paper Topic And Main Contributions:**

This work presents a novel method, SUPMER, that incorporates meta-learning concepts to improve the prompting of pre-trained language models. The different method components tackle common problems of existing prompting methods, namely the need for a good soft prompt initialization and overfitting in few-shot settings due to the small number of training examples. The results indicate the superiority of SUPMER across a variety of tasks.

**Questions For The Authors:**

- How exactly are the sentence embeddings computed for the k-means clustering?

**Reasons To Accept:**

- Strong empirical evidence highlighting the superiority of the method in few-shot learning and domain generalization experiments.
- The paper is well written, despite all the complexity and moving parts of the proposed method.
- The work focuses on the relevant problem of prompting pre-trained language models, an efficient and practical alternative to finetuning.

**Reasons To Reject:**

- The experiments are performed on one model family: T5. Particularly, T5-base and Flan-T5-XL, which limits the scope of the work. However, the method seems to distance itself from other baselines with the larger model, which is a good indication of being scale-friendly.

**Reproducibility:**

3: Could reproduce the results with some difficulty. The settings of parameters are underspecified or subjectively determined; the training/evaluation data are not widely available.

**Reviewer Confidence:**

2: Willing to defend my evaluation, but it is fairly likely that I missed some details, didn't understand some central points, or can't be sure about the novelty of the work.

---

> ### Author Rebuttal · Authors · 2023-08-29
>
> We sincerely thank you for the valuable comments! We are encouraged that you think our work presents a **novel method with strong empirical evidence highlighting its superiority**. We will explain your concerns point by point.
>
> &nbsp;
>
> > **[1]** The current experiments are performed on one model family: T5.
>
> To further demonstrate that SUPMER can also achieve superior performance on LMs beyond the T5 family, we newly leverage **GPT-2-Large** as the backbone model and conduct experiments under the same experimental setup. We compare SUPMER with FT, PT, PPT, and MetaPT.
>
> The experimental results are shown in the following tables. We can see that **SUPMER outperforms all baselines** across 10 of all 12 datasets in few-shot learning, and **demonstrates superior performance** than all baselines across 5 of all 6 domains in domain generalization. **SUPMER also achieves significantly better average performance** in terms of both few-shot learning and domain generalization. So we can say that **SUPMER still demonstrates its effectiveness on LMs beyond the T5 family**.
>
> &nbsp;
>
> **Experimental results of few-shot learning on GPT-2-Large**:
> | **Method & Dataset** | SST-2 | SST-5 | MR | CR | SUBJ | TREC | CB | RTE | QNLI | WiC | MRPC | QQP | *AVG* |
> | :--- | :---: | :---: | :---: | :---: | :---: | :---: | :---: | :---: | :---: | :---: | :---: | :---: | :---: |
> | FT | 84.0 | 41.5 | 81.4 | 83.1 | **85.3** | **79.0** | 69.7 | 60.2 | 62.7 | 54.4 | 70.1 | 71.0 | 70.2 |
> | PT | 73.2 | 32.4 | 75.6 | 79.8 | 65.7 | 64.2 | 62.9 | 57.0 | 59.5 | 53.2 | 67.8 | 66.6 | 63.2 |
> | PPT | 79.7 | 34.1 | 79.3 | 82.3 | 72.3 | 71.6 | 66.1 | 59.3 | 62.2 | 53.5 | 69.4 | 67.0 | 66.4 |
> | MetaPT | 85.1 | 42.8 | 84.7 | 87.0 | 77.4 | 75.6 | 67.9 | 58.5 | 61.4 | 53.9 | 69.2 | 70.2 | 69.5 |
> | SUPMER | **86.2** | **45.2** | **85.3** | **87.8** | 84.6 | 78.4 | **71.4** | **60.8** | **64.2** | **54.9** | **70.6** | **72.8** | **71.9** |
>
> **Experimental results of domain generalization on GPT-2-Large**:
> | **Method & Domain** | B | D | E | K | A | R | *AVG_SUB* | *AVG* |
> | :--- | :---: | :---: | :---: | :---: | :---: | :---: | :---: | :---: |
> | FT | 84.6 | 83.5 | 81.7 | 84.0 | 86.6 | 85.8 | 84.1 | 84.4 |
> | PT | 80.7 | 76.9 | 77.2 | 79.0 | 79.8 | 83.1 | 78.7 | 79.5 |
> | PPT | 83.2 | 79.7 | 83.7 | 88.3 | 85.3 | 85.9 | 84.0 | 84.4 |
> | MetaPT | **85.7** | 85.1 | 85.0 | 88.8 | 90.7 | --- | 87.1 | --- |
> | SUPMER | 85.3 | **86.4** | **86.3** | **90.4** | **91.9** | **88.8** | **88.1** | **88.2** |
>
> &nbsp;
>
> > **[2]** How exactly are the sentence embeddings computed for the k-means clustering?
>
> To derive the sentence embedding for each sentence from the unlabeled corpora, we leverage the pre-trained T5 encoder for sentence encoding. The hidden representations from the last layer of the T5 encoder are extracted, and we subsequently employ mean pooling to aggregate the token embeddings, thereby generating the sentence embedding. With sentence embeddings, then we can directly apply K-means to cluster the unlabeled sentences. We give more details about this in Appendix A.2 within our paper.
>
> &nbsp;
>
> > **[3]** Typos Grammar Style And Presentation Improvements: line 168 -> space missing
>
> Thank you for pointing out the omission! We will add the missing space in line 168 in the next version. Additionally, we will also incorporate the aforementioned experiments using GPT-2-large as the backbone in the next version. Once again, we sincerely appreciate your constructive advice!

---

### Official Review · Reviewer_33e1 · 2023-08-07

**Soundness:** 4

**Excitement:**

4: Strong: This paper deepens the understanding of some phenomenon or lowers the barriers to an existing research direction.

**Paper Topic And Main Contributions:**

The author presents SUPMER, a self-supervised algorithm that uses a set of anchor tasks within one or more datasets to use meta-gradient regularization to learn meta-prompts which are then used to train on a series of meta-tasks. By incorporating meta-learning, the authors have been able to create a novel method that learns new data over n-shots by learning the task within this meta-embedding.

**Questions For The Authors:**

I see no major issues with the work. Given other research and works presented on the topic of few-shot learning, it seems that the results shown fall in line with expectations given other similar works. The authors appear to provide a novel method that advances previous works.

**Reasons To Accept:**

The author outlines their novel method thoroughly, breaking the problem down into distinct steps.

The paper provides a rich comparison between their method and other existing methods. They give results for two models of different sizes, across multiple datasets, and with different tasks. Additionally, they outline the limitations of their study and discuss possible future investigations.

The author provides the results of an ablation study to show that each component of their method is contributing to the overall advantage in the performance of their method.

**Reasons To Reject:**

Figure 2 could be broken down and better explained as separate figures, which could then be combined into what we see in Figure 2. The meaning behind the elements of your figure is not always immediately clear. For example, the meaning of the colored dots in Task Interpolation becomes clear after some analysis of the text and diagram but it also would be better served by an explanation specifically referring to the Curriculum-based Task Augmentation part of the diagram within Figure 2.

The introduction could be revised to better introduce key concepts before beginning to discuss the motivations of the paper. This is not critical as the authors always define a term before discussing it. However, the introduction could more elegantly describe the base problem and all relevant terms before then moving to discuss the more specific motivations that lead to the methods which are presented in this paper.

**Reproducibility:**

4: Could mostly reproduce the results, but there may be some variation because of sample variance or minor variations in their interpretation of the protocol or method.

**Reviewer Confidence:**

3: Pretty sure, but there's a chance I missed something. Although I have a good feel for this area in general, I did not carefully check the paper's details, e.g., the math, experimental design, or novelty.

---

> ### Author Rebuttal · Authors · 2023-08-29
>
> We sincerely appreciate your recognition of our work and the valuable suggestions you've provided! We are encouraged to see that our work is recognized as **novel, backed by rich experiments**. And we will diligently adopt your suggestions on presentation improvement and revise Figure 2 and the section of introduction in the next version.
>
> Based on your suggestions, we will divide Figure 2 into separate figures for better explanation, while also incorporating some essential textual annotations. We will make a clearer exposition of various elements within the figure, such as the task interpolation and the curriculum-based approach. Furthermore, within the introduction section, we will endeavor to provide a more elegant depiction of the base problem and all relevant terms, ensuring a more intuitive presentation for readers.
>
> Once again, we extend our heartfelt gratitude to you!

---

### Official Review · Reviewer_a2Py · 2023-08-12

**Typos Grammar Style And Presentation Improvements:** 1. Figure 2 is quite difficult to und…
**Soundness:** 4

**Excitement:**

3: Ambivalent: It has merits (e.g., it reports state-of-the-art results, the idea is nice), but there are key weaknesses (e.g., it describes incremental work), and it can significantly benefit from another round of revision. However, I won't object to accepting it if my co-reviewers champion it.

**Paper Topic And Main Contributions:**

This work introduces an approach that utilizes meta-learning to effectively train a model with soft-prompt tuning. The method proposes a self-supervised framework to initialize the soft-prompts, integrate a regularization that helps with domain-generalization. The method is compared against existing approaches such as full-model finetuning.

**Reasons To Accept:**

1. Novel approach to initialize soft-prompt tuning which also works in a self-supervised paradigm, meaning there is virtually no need for a human-in-the-loop.
2. Shows that the results stay consistently high as training continues as oppose to other approaches that deteriorate over time.

**Reasons To Reject:**

1. While the approach is interesting, the evaluation done is less compelling given the model used in this work, specifically Flan-T5-XL. The model is based on T5-v1.1 models finetuned on a collection of dataset which already includes TREC and QQP. It is therefore still unclear if the difference between finetuning and SUPMER is due to overfitting on FT.
2. FLAN-T5-XL was also designed with few-shot inference in mind which may have contributed to the higher gains when compared to a model that wasn't such as T5-Base (Although size may very well be a factor). This means there are a few unobserved variables that were taken to account.
3. To effectively prove SUPMER's efficacy on FLAN-T5-XL, comparing SUPMER with few-shot inference and showing that SUPMER is still better despite FLAN-T5-XL's initial design would have even shown how important this method is. Instead of FLAN-T5-XL, had this work use models such as T5-v1.1 which was only trained on C4, the results may have become more convincing.

**Reproducibility:**

3: Could reproduce the results with some difficulty. The settings of parameters are underspecified or subjectively determined; the training/evaluation data are not widely available.

**Reviewer Confidence:**

3: Pretty sure, but there's a chance I missed something. Although I have a good feel for this area in general, I did not carefully check the paper's details, e.g., the math, experimental design, or novelty.

---

> ### Author Rebuttal · Authors · 2023-08-29
>
> We sincerely thank you for the valuable comments! We are encouraged to see that our work is recognized as **novel and interesting**. To better explain your concerns, we have added some new experiments:
> 1. Comparing with **few-shot inference on Flan-T5-XL**.
> 2. Leveraging **T5-v1.1-XL** as backbone model.
> 3. Leveraging **GPT-2-Large** as backbone model.
>
> &nbsp;
>
> ### **1. Comparing with few-shot inference on Flan-T5-XL**.
> Given that  FLAN-T5-XL was also designed with few-shot inference in mind, we newly compare with two baseline methods on FLAN-T5-XL: **zero-shot inference** and **few-shot inference**. For both of them, we directly employ Flan-T5-XL for downstream evaluation, coupled with carefully designed task instructions for each dataset. Furthermore, in few-shot inference, we also provide an appropriate number of few-shot examples to form a demonstration context. The experimental results are as follows:
>
>
> **Experimental results of few-shot learning on FlanT5-XL:**
> | **Method & Dataset** | SST-2 | SST-5 | MR | CR | SUBJ | TREC | CB | RTE | QNLI | WiC | MRPC | QQP | *AVG* |
> | :--- | :---: | :---: | :---: | :---: | :---: | :---: | :---: | :---: | :---: | :---: | :---: | :---: | :---: |
> | zero-shot inference | 89.1 | 52.3 | 83.3 | 80.6 | 57.4 | 87.2 | 76.8 | 75.8 | 85.0 | 50.5 | 77.2 | 77.5 | 74.4 |
> | few-shot inference | 93.2 | 53.3 | 88.5 | 87.8 | 58.6 | 91.6 | 83.9 | 79.1 | 86.9 | 64.3 | **79.9** | 81.0 | 79.0 |
> | FT | 92.9 | 53.6 | 89.6 | **91.3** | 88.7 | **93.6** | 77.8 | 76.0 | 84.4 | 63.0 | 77.7 | 82.9 | 81.0 |
> | PT | 88.2 | 45.7 | 85.6 | 88.6 | 81.6 | 85.1 | 65.4 | 61.6 | 69.3 | 54.8 | 71.1 | 70.2 | 72.3 |
> | SUPMER | **95.5** | **55.3** | **91.4** | 90.7 | **90.3** | 93.0 | **87.6** | **81.4** | **88.3** | **65.0** | 78.1 | **85.1** | **83.5** |
>
> **Experimental results of domain generalization on FlanT5-XL:**
> | **Method & Domain** | B | D | E | K | A | R | *AVG* |
> | :--- | :---: | :---: | :---: | :---: | :---: | :---: | :---: |
> | zero-shot inference | 77.8 | 84.6 | 86.2 | 86.8 | 88.6 | 87.8 | 85.3 |
> | few-shot inference | 82.5 | 90.3 | 89.7 | 92.3 | 92.2 | 89.2 | 89.4 |
> | FT | 86.2 | 88.8 | 84.6 | 87.2 | 89.8 | 90.7 | 87.9 |
> | PT | 82.6 | 79.0 | 82.5 | 84.2 | 84.5 | 84.8 | 82.9 |
> | SUPMER | **87.0** | **89.8** | **91.1** | **95.1** | **95.8** | **91.8** |  **91.8** |
>
> > **[1.1]**  Whether the difference between fine-tuning and SUPMER is due to overfitting on FT?
>
> First, we can see that few-shot inference surpasses FT on some datasets. This validates that for FLAN-T5-XL, employing few-shot examples as the demonstration context during inference, rather than utilizing them for parameter tuning, can also showcase impressive capabilities. Thanks for your constructive suggestion, and ***it is indeed necessary to consider few-shot inference as a baseline method*** for comparative analysis.
>
> Furthermore, comparing the results of zero-shot inference and FT, we can see that after fine-tuning model parameters based on few-shot examples, **FT consistently outperforms zero-shot inference across almost all datasets, including QQP and TREC** (FT also achieves better average performance than few-shot inference in the experiments of few-shot learning). During inference, both zero-shot inference and FT only utilize test samples without the incorporation of any additional few-shot in-context examples. The superior performance achieved by FT compared to zero-shot inference, thereby underscores ***the positive impact of FT on enhancing model performance, without encountering overfitting*** which may lead to a decline in performance.
>
> > **[1.2]**  Comparing SUPMER with few-shot inference
>
> For both few-shot learning and domain generalization, **SUPMER demonstrates superior performance across almost all datasets and domains in contrast to few-shot inference**. Specifically, SUPMER exhibits average gains of **4.5** points in few-shot learning and **2.4** points in domain generalization, compared to few-shot inference.  These results provide further evidence that ***for LMs such as FLAN-T5XL with inherent few-shot inference capabilities, our approach can significantly enhance their abilities in a parameter-efficient tuning strategy***, without providing any in-context examples during inference. This effectively proves SUPMER's efficacy on FLAN-T5-XL.
>
> &nbsp;
>
> ### **2. Leveraging T5-v1.1-XL as backbone model**
>
> As FLAN-T5-XL was fine-tuned on a collection of datasets with the ability of few-shot inference in mind, we employ **T5-v1.1-XL** as the backbone model to ensure a more fair comparison between FT and SUPMER. T5-v1.1-XL was only pre-trained on the unsupervised dataset C4, so the primary distinction between T5-v1.1-XL and T5-base lies in their model scales.
>
> We list the experimental results on T5-v1.1-XL in the tables below. We have the following observations:
>
> (1) **When utilizing T5-v1.1-XL as the backbone, SUPMER still significantly outperforms FT**, exhibiting average gains of 3.1 points in few-shot learning and 5.5 points in domain generalization. This underscores the effectiveness of SUPMER. Furthermore, in comparison to the experimental results obtained by employing T5-base as the backbone model (refer to Tables 1 and 2 of the main paper), the performance improvement of SUPMER over other baseline methods is more significant when utilizing the larger T5-v1.1-XL. ***It shows that SUPMER has a good indication of being scale-friendly.***
>
> (2) Upon integrating SUPMER with T5-v1.1-XL, its performance in both few-shot learning and domain generation surpasses the zero-shot performance of Flan-T5-XL. This underscores that, during inference without few-shot examples, *by additionally integrating the ingeniously trained soft prompts from our approach into the LM input,* ***even models that have not undergone instruction-tuning, can still achieve superior performance compared to their instruction-tuned counterparts***.
>
> &nbsp;
>
> **Experimental results of few-shot learning on T5-v1.1-XL:**
> | **Method & Dataset** | SST-2 | SST-5 | MR | CR | SUBJ | TREC | CB | RTE | QNLI | WiC | MRPC | QQP | *AVG* |
> | :--- | :---: | :---: | :---: | :---: | :---: | :---: | :---: | :---: | :---: | :---: | :---: | :---: | :---: |
> | FT | 90.8 | 42.8 | 86.2 | 88.6 | 84.5 | **88.6** | 76.8 | 58.5 | 66.8 | 54.5 | 70.4 | 70.7 | 73.3 |
> | PT | 73.6 | 39.5 | 80.0 | 86.9 | 73.7 | 75.4 | 60.7 | 55.2 | 61.6 | 53.3 | 68.4 | 67.8 | 66.3 |
> | SUPMER | **92.0** | **46.9** | **88.0** | **89.4** | **89.5** | 88.2 | **85.7** | **60.4** | **70.8** | **57.9** | **73.2** | **74.3** | **76.4** |
>
> **Experimental results of domain generalization on T5-v1.1-XL:**
> | **Method & Domain** | B | D | E | K | A | R | *AVG* |
> | :--- | :---: | :---: | :---: | :---: | :---: | :---: | :---: |
> | FT | 85.8 | 84.2 | 82.6 | 85.2 | 85.7 | 85.5 | 84.8 |
> | PT | 80.0 | 77.5 | 80.1 | 82.8 | 83.4 | 83.4 | 81.2 |
> | SUPMER | **86.7** | **88.1** | **88.7** | **93.2** | **94.1** | **91.0** | **90.3** |
>
> &nbsp;
>
> ### **3. Leveraging **GPT-2-Large** as backbone model**
>
> In addition to the backbone model from the T5 model family, we have also conducted experiments on GPT-2-Large. We compare SUPMER with FT, PT, PPT, and MetaPT. The experimental results are listed in the following tables. We can see that ***SUPMER can still achieve better performance on LMs beyond the T5 family.*** Specifically, SUPMER outperforms all baselines across 10 of all 12 datasets in few-shot learning, and demonstrates superior performance than all baselines across 5 of all 6 domains in domain generalization, while also achieving significantly better average performance.
>
> &nbsp;
>
> **Experimental results of few-shot learning on GPT-2-Large:**
> | **Method & Dataset** | SST-2 | SST-5 | MR | CR | SUBJ | TREC | CB | RTE | QNLI | WiC | MRPC | QQP | *AVG* |
> | :--- | :---: | :---: | :---: | :---: | :---: | :---: | :---: | :---: | :---: | :---: | :---: | :---: | :---: |
> | FT | 84.0 | 41.5 | 81.4 | 83.1 | **85.3** | **79.0** | 69.7 | 60.2 | 62.7 | 54.4 | 70.1 | 71.0 | 70.2 |
> | PT | 73.2 | 32.4 | 75.6 | 79.8 | 65.7 | 64.2 | 62.9 | 57.0 | 59.5 | 53.2 | 67.8 | 66.6 | 63.2 |
> | PPT | 79.7 | 34.1 | 79.3 | 82.3 | 72.3 | 71.6 | 66.1 | 59.3 | 62.2 | 53.5 | 69.4 | 67.0 | 66.4 |
> | MetaPT | 85.1 | 42.8 | 84.7 | 87.0 | 77.4 | 75.6 | 67.9 | 58.5 | 61.4 | 53.9 | 69.2 | 70.2 | 69.5 |
> | SUPMER | **86.2** | **45.2** | **85.3** | **87.8** | 84.6 | 78.4 | **71.4** | **60.8** | **64.2** | **54.9** | **70.6** | **72.8** | **71.9** |
>
> **Experimental results of domain generalization on GPT-2-Large:**
> | **Method & Domain** | B | D | E | K | A | R | *AVG_SUB* | *AVG* |
> | :--- | :---: | :---: | :---: | :---: | :---: | :---: | :---: | :---: |
> | FT | 84.6 | 83.5 | 81.7 | 84.0 | 86.6 | 85.8 | 84.1 | 84.4 |
> | PT | 80.7 | 76.9 | 77.2 | 79.0 | 79.8 | 83.1 | 78.7 | 79.5 |
> | PPT | 83.2 | 79.7 | 83.7 | 88.3 | 85.3 | 85.9 | 84.0 | 84.4 |
> | MetaPT | **85.7** | 85.1 | 85.0 | 88.8 | 90.7 | --- | 87.1 | --- |
> | SUPMER | 85.3 | **86.4** | **86.3** | **90.4** | **91.9** | **88.8** | **88.1** | **88.2** |
>
> &nbsp;
>
> Overall, we extend our heartfelt gratitude for your valuable suggestions! The next version will encompass the aforementioned experiments. We will also thoroughly incorporate your suggestions on presentation improvement and make revisions to Figure 2 in the next version.

---

### Meta-Review · Area_Chair_pXvW · 2023-09-07

**Recommendation:** 4

**Metareview:**

This paper introduces SUPMER which leverages meta-learning to enhance the prompting of PLMs. SUPMER utilizes a self-supervised framework to initialize soft-prompts and incorporates meta-gradient regularization, enabling domain generalization. Compared to existing methods, SUPMER demonstrates superior performance by learning meta-prompts through anchor tasks and applying them to a series of meta-tasks. SUPMER addresses common challenges in prompting methods, such as the need for effective soft prompt initialization and mitigating overfitting in few-shot settings due to limited training examples.


The reviewers initially raised many concerns and the authors provided extensive rebuttal responses. All reviewers engaged in post-rebuttal discussions and converged to find the work sound with moderate excitement scores and acceptable reproducibility.

---

### Decision · Program_Chairs · 2023-10-07

**Decision:**

Accept-Findings

**Comment:**

This paper introduces SUPMER which leverages meta-learning to enhance the prompting of PLMs. SUPMER utilizes a self-supervised framework to initialize soft-prompts and incorporates meta-gradient regularization, enabling domain generalization. Compared to existing methods, SUPMER demonstrates superior performance by learning meta-prompts through anchor tasks and applying them to a series of meta-tasks. SUPMER addresses common challenges in prompting methods, such as the need for effective soft prompt initialization and mitigating overfitting in few-shot settings due to limited training examples.


The reviewers initially raised many concerns and the authors provided extensive rebuttal responses. All reviewers engaged in post-rebuttal discussions and converged to find the work sound with moderate excitement scores and acceptable reproducibility.